# SELF-TUNING NETWORKS: BILEVEL OPTIMIZATION OF HYPERPARAMETERS USING STRUCTURED BEST-RESPONSE FUNCTIONS

**Matthew MacKay**[∗]**, Paul Vicol**[∗]**, Jon Lorraine, David Duvenaud, Roger Grosse**

{mmackay,pvicol,lorraine,duvenaud,rgrosse}@cs.toronto.edu

University of Toronto

Vector Institute

## ABSTRACT

Hyperparameter optimization can be formulated as a bilevel optimization problem, where the optimal parameters on the training set depend on the hyperparameters. We aim to adapt regularization hyperparameters for neural networks by fitting compact approximations to the *best-response function*, which maps hyperparameters to optimal weights and biases. We show how to construct scalable best-response approximations for neural networks by modeling the best-response as a single network whose hidden units are gated conditionally on the regularizer. We justify this approximation by showing the exact best-response for a shallow linear network with $L_2$-regularized Jacobian can be represented by a similar gating mechanism. We fit this model using a gradient-based hyperparameter optimization algorithm which alternates between approximating the best-response around the current hyperparameters and optimizing the hyperparameters using the approximate best-response function. Unlike other gradient-based approaches, we do not require differentiating the training loss with respect to the hyperparameters, allowing us to tune discrete hyperparameters, data augmentation hyperparameters, and dropout probabilities. Because the hyperparameters are adapted online, our approach discovers *hyperparameter schedules* that can outperform fixed hyperparameter values. Empirically, our approach outperforms competing hyperparameter optimization methods on large-scale deep learning problems. We call our networks, which update their own hyperparameters online during training, *Self-Tuning Networks (STNs)*.

## 1 INTRODUCTION

Regularization hyperparameters such as weight decay, data augmentation, and dropout (Srivastava et al., 2014) are crucial to the generalization of neural networks, but are difficult to tune. Popular approaches to hyperparameter optimization include grid search, random search (Bergstra & Bengio, 2012), and Bayesian optimization (Snoek et al., 2012). These approaches work well with low-dimensional hyperparameter spaces and ample computational resources; however, they pose hyperparameter optimization as a black-box optimization problem, ignoring structure which can be exploited for faster convergence, and require many training runs.

We can formulate hyperparameter optimization as a bilevel optimization problem. Let $\mathbf{w}$ denote parameters (e.g. weights and biases) and $\boldsymbol{\lambda}$ denote hyperparameters (e.g. dropout probability). Let $\mathcal{L}_T$ and $\mathcal{L}_V$ be functions mapping parameters and hyperparameters to training and validation losses, respectively. We aim to solve[1]:

$$\boldsymbol{\lambda}^* = \arg\min_{\boldsymbol{\lambda}} \mathcal{L}_V(\boldsymbol{\lambda}, \mathbf{w}^*) \quad \text{subject to} \quad \mathbf{w}^* = \arg\min_{\mathbf{w}} \mathcal{L}_T(\boldsymbol{\lambda}, \mathbf{w}) \tag{1}$$

---

[∗]Equal contribution.

[1]The uniqueness of the $\arg\min$ is assumed.

Substituting the *best-response function* $\mathbf{w}^*(\boldsymbol{\lambda}) = \arg\min_{\mathbf{w}} \mathcal{L}_T(\boldsymbol{\lambda}, \mathbf{w})$ gives a single-level problem:

$$\boldsymbol{\lambda}^* = \arg\min_{\boldsymbol{\lambda}} \mathcal{L}_V(\boldsymbol{\lambda}, \mathbf{w}^*(\boldsymbol{\lambda})) \tag{2}$$

If the best-response $\mathbf{w}^*$ is known, the validation loss can be minimized directly by gradient descent using Equation 2, offering dramatic speed-ups over black-box methods. However, as the solution to a high-dimensional optimization problem, it is difficult to compute $\mathbf{w}^*$ even approximately.

Following Lorraine & Duvenaud (2018), we propose to approximate the best-response $\mathbf{w}^*$ directly with a parametric function $\hat{\mathbf{w}}_\phi$. We jointly optimize $\phi$ and $\boldsymbol{\lambda}$, first updating $\phi$ so that $\hat{\mathbf{w}}_\phi \approx \mathbf{w}^*$ in a neighborhood around the current hyperparameters, then updating $\boldsymbol{\lambda}$ by using $\hat{\mathbf{w}}_\phi$ as a proxy for $\mathbf{w}^*$ in Eq. 2:

$$\boldsymbol{\lambda}^* \approx \arg\min_{\boldsymbol{\lambda}} \mathcal{L}_V(\boldsymbol{\lambda}, \hat{\mathbf{w}}_\phi(\boldsymbol{\lambda})) \tag{3}$$

Finding a scalable approximation $\hat{\mathbf{w}}_\phi$ when $\mathbf{w}$ represents the weights of a neural network is a significant challenge, as even simple implementations entail significant memory overhead. We show how to construct a compact approximation by modelling the best-response of each row in a layer's weight matrix/bias as a rank-one affine transformation of the hyperparameters. We show that this can be interpreted as computing the activations of a base network in the usual fashion, plus a correction term dependent on the hyperparameters. We justify this approximation by showing the exact best-response for a shallow linear network with $L_2$-regularized Jacobian follows a similar structure. We call our proposed networks *Self-Tuning Networks* (STNs) since they update their own hyperparameters online during training.

STNs enjoy many advantages over other hyperparameter optimization methods. First, they are easy to implement by replacing existing modules in deep learning libraries with "hyper" counterparts which accept an additional vector of hyperparameters as input[2]. Second, because the hyperparameters are adapted online, we ensure that computational effort expended to fit $\phi$ around previous hyperparameters is not wasted. In addition, this online adaption yields hyperparameter schedules which we find empirically to outperform fixed hyperparameter settings. Finally, the STN training algorithm does not require differentiating the training loss with respect to the hyperparameters, unlike other gradient-based approaches (Maclaurin et al., 2015; Larsen et al., 1996), allowing us to tune discrete hyperparameters, such as the number of holes to cut out of an image (DeVries & Taylor, 2017), data-augmentation hyperparameters, and discrete-noise dropout parameters. Empirically, we evaluate the performance of STNs on large-scale deep-learning problems with the Penn Treebank (Marcus et al., 1993) and CIFAR-10 datasets (Krizhevsky & Hinton, 2009), and find that they substantially outperform baseline methods.

## 2 BILEVEL OPTIMIZATION

A bilevel optimization problem consists of two sub-problems called the *upper-level* and *lower-level* problems, where the upper-level problem must be solved subject to optimality of the lower-level problem. Minimax problems are an example of bilevel programs where the upper-level objective equals the negative lower-level objective. Bilevel programs were first studied in economics to model leader/follower firm dynamics (Von Stackelberg, 2010) and have since found uses in various fields (see Colson et al. (2007) for an overview). In machine learning, many problems can be formulated as bilevel programs, including hyperparameter optimization, GAN training (Goodfellow et al., 2014), meta-learning, and neural architecture search (Zoph & Le, 2016).

Even if all objectives and constraints are linear, bilevel problems are strongly NP-hard (Hansen et al., 1992; Vicente et al., 1994). Due to the difficulty of obtaining exact solutions, most work has focused on restricted settings, considering linear, quadratic, and convex functions. In contrast, we focus on obtaining local solutions in the nonconvex, differentiable, and unconstrained setting. Let $F, f : \mathbb{R}^n \times \mathbb{R}^m \to \mathbb{R}$ denote the upper- and lower-level objectives (e.g., $\mathcal{L}_V$ and $\mathcal{L}_T$) and $\boldsymbol{\lambda} \in \mathbb{R}^n, \mathbf{w} \in \mathbb{R}^m$ denote the upper- and lower-level parameters. We aim to solve:

$$\min_{\boldsymbol{\lambda} \in \mathbb{R}^n} F(\boldsymbol{\lambda}, \mathbf{w}) \tag{4a}$$

$$\text{subject to} \quad \mathbf{w} \in \arg\min_{\mathbf{w} \in \mathbb{R}^m} f(\boldsymbol{\lambda}, \mathbf{w}) \tag{4b}$$

---

[2]We illustrate how this is done for the PyTorch library (Paszke et al., 2017) in Appendix G.

It is desirable to design a gradient-based algorithm for solving Problem 4, since using gradient information provides drastic speed-ups over black-box optimization methods (Nesterov, 2013). The simplest method is simultaneous gradient descent, which updates $\boldsymbol{\lambda}$ using $\partial F/\partial\boldsymbol{\lambda}$ and $\mathbf{w}$ using $\partial f/\partial\mathbf{w}$. However, simultaneous gradient descent often gives incorrect solutions as it fails to account for the dependence of $\mathbf{w}$ on $\boldsymbol{\lambda}$. Consider the relatively common situation where $F$ doesn't depend directly on $\boldsymbol{\lambda}$, so that $\partial F/\partial\boldsymbol{\lambda} \equiv \mathbf{0}$ and hence $\boldsymbol{\lambda}$ is never updated.

## 2.1 GRADIENT DESCENT VIA THE BEST-RESPONSE FUNCTION

A more principled approach to solving Problem 4 is to use the *best-response function* (Gibbons, 1992). Assume the lower-level Problem 4b has a unique optimum $\mathbf{w}^*(\boldsymbol{\lambda})$ for each $\boldsymbol{\lambda}$. Substituting the best-response function $\mathbf{w}^*$ converts Problem 4 into a single-level problem:

$$\min_{\boldsymbol{\lambda}\in\mathbb{R}^n} F^*(\boldsymbol{\lambda}) := F(\boldsymbol{\lambda}, \mathbf{w}^*(\boldsymbol{\lambda})) \tag{5}$$

If $\mathbf{w}^*$ is differentiable, we can minimize Eq. 5 using gradient descent on $F^*$ with respect to $\boldsymbol{\lambda}$. This method requires a unique optimum $\mathbf{w}^*(\boldsymbol{\lambda})$ for Problem 4b for each $\boldsymbol{\lambda}$ and differentiability of $\mathbf{w}^*$. In general, these conditions are difficult to verify. We give sufficient conditions for them to hold in a neighborhood of a point $(\boldsymbol{\lambda}_0, \mathbf{w}_0)$ where $\mathbf{w}_0$ solves Problem 4b given $\boldsymbol{\lambda}_0$.

**Lemma 1.** *(Fiacco & Ishizuka, 1990) Let $\boldsymbol{w}_0$ solve Problem 4b for $\boldsymbol{\lambda}_0$. Suppose $f$ is $\mathcal{C}^2$ in a neighborhood of $(\boldsymbol{\lambda}_0, \boldsymbol{w}_0)$ and the Hessian $\partial^2 f/\partial w^2(\boldsymbol{\lambda}_0, \boldsymbol{w}_0)$ is positive definite. Then for some neighborhood $U$ of $\boldsymbol{\lambda}_0$, there exists a continuously differentiable function $\boldsymbol{w}^* : U \to \mathbb{R}^m$ such that $\boldsymbol{w}^*(\boldsymbol{\lambda})$ is the unique solution to Problem 4b for each $\boldsymbol{\lambda} \in U$ and $\boldsymbol{w}^*(\boldsymbol{\lambda}_0) = \boldsymbol{w}_0$.*

*Proof.* See Appendix B.1. $\square$

The gradient of $F^*$ decomposes into two terms, which we term the *direct gradient* and the *response gradient*. The direct gradient captures the direct reliance of the upper-level objective on $\boldsymbol{\lambda}$, while the response gradient captures how the lower-level parameter responds to changes in the upper-level parameter:

$$\frac{\partial F^*}{\partial\boldsymbol{\lambda}}(\boldsymbol{\lambda}_0) = \underbrace{\frac{\partial F}{\partial\boldsymbol{\lambda}}(\boldsymbol{\lambda}_0, \mathbf{w}^*(\boldsymbol{\lambda}_0))}_{\text{Direct gradient}} + \underbrace{\frac{\partial F}{\partial\mathbf{w}}(\boldsymbol{\lambda}_0, \mathbf{w}^*(\boldsymbol{\lambda}_0))\frac{\partial\mathbf{w}^*}{\partial\boldsymbol{\lambda}}(\boldsymbol{\lambda}_0)}_{\text{Response gradient}} \tag{6}$$

Even if $\partial F/\partial\boldsymbol{\lambda} \not\equiv 0$ and simultaneous gradient descent is possible, including the response gradient can stabilize optimization by converting the bilevel problem into a single-level one, as noted by Metz et al. (2016) for GAN optimization. Conversion to a single-level problem ensures that the gradient vector field is conservative, avoiding pathological issues described by Mescheder et al. (2017).

## 2.2 APPROXIMATING THE BEST-RESPONSE FUNCTION

In general, the solution to Problem 4b is a set, but assuming uniqueness of a solution and differentiability of $\mathbf{w}^*$ can yield fruitful algorithms in practice. In fact, gradient-based hyperparameter optimization methods can often be interpreted as approximating either the best-response $\mathbf{w}^*$ or its Jacobian $\partial\mathbf{w}^*/\partial\boldsymbol{\lambda}$, as detailed in Section 5. However, these approaches can be computationally expensive and often struggle with discrete hyperparameters and stochastic hyperparameters like dropout probabilities, since they require differentiating the training loss with respect to the hyperparameters. Promising approaches to approximate $\mathbf{w}^*$ directly were proposed by Lorraine & Duvenaud (2018), and are detailed below.

**1. Global Approximation.** The first algorithm proposed by Lorraine & Duvenaud (2018) approximates $\mathbf{w}^*$ as a differentiable function $\hat{\mathbf{w}}_\phi$ with parameters $\phi$. If $\mathbf{w}$ represents neural net weights, then the mapping $\hat{\mathbf{w}}_\phi$ is a hypernetwork (Schmidhuber, 1992; Ha et al., 2016). If the distribution $p(\boldsymbol{\lambda})$ is fixed, then gradient descent with respect to $\phi$ minimizes:

$$\mathbb{E}_{\boldsymbol{\lambda}\sim p(\boldsymbol{\lambda})}\left[f(\boldsymbol{\lambda}, \hat{\mathbf{w}}_\phi(\boldsymbol{\lambda}))\right] \tag{7}$$

If support$(p)$ is broad and $\hat{\mathbf{w}}_\phi$ is sufficiently flexible, then $\hat{\mathbf{w}}_\phi$ can be used as a proxy for $\mathbf{w}^*$ in Problem 5, resulting in the following objective:

$$\min_{\boldsymbol{\lambda}\in\mathbb{R}^n} F(\boldsymbol{\lambda}, \hat{\mathbf{w}}_\phi(\boldsymbol{\lambda})) \tag{8}$$

**2. Local Approximation.** In practice, $\hat{\mathbf{w}}_\phi$ is usually insufficiently flexible to model $\mathbf{w}^*$ on support($p$). The second algorithm of Lorraine & Duvenaud (2018) locally approximates $\mathbf{w}^*$ in a neighborhood around the current upper-level parameter $\boldsymbol{\lambda}$. They set $p(\boldsymbol{\epsilon}|\boldsymbol{\sigma})$ to a factorized Gaussian noise distribution with a fixed scale parameter $\boldsymbol{\sigma} \in \mathbb{R}^n_+$, and found $\phi$ by minimizing the objective:

$$\mathbb{E}_{\boldsymbol{\epsilon} \sim p(\boldsymbol{\epsilon}|\boldsymbol{\sigma})} \left[ f(\boldsymbol{\lambda} + \boldsymbol{\epsilon}, \hat{\mathbf{w}}_\phi(\boldsymbol{\lambda} + \boldsymbol{\epsilon})) \right] \tag{9}$$

Intuitively, the upper-level parameter $\boldsymbol{\lambda}$ is perturbed by a small amount, so the lower-level parameter learns how to respond. An alternating gradient descent scheme is used, where $\phi$ is updated to minimize equation 9 and $\boldsymbol{\lambda}$ is updated to minimize equation 8. This approach worked for problems using $L_2$ regularization on MNIST (LeCun et al., 1998). However, it is unclear if the approach works with different regularizers or scales to larger problems. It requires $\hat{\mathbf{w}}_\phi$, which is *a priori* unwieldy for high dimensional $\mathbf{w}$. It is also unclear how to set $\boldsymbol{\sigma}$, which defines the size of the neighborhood on which $\phi$ is trained, or if the approach can be adapted to discrete and stochastic hyperparameters.

## 3 SELF-TUNING NETWORKS

In this section, we first construct a best-response approximation $\hat{\mathbf{w}}_\phi$ that is memory efficient and scales to large neural networks. We justify this approximation through analysis of simpler situations. Then, we describe a method to automatically adjust the scale of the neighborhood $\phi$ is trained on. Finally, we formally describe our algorithm and discuss how it easily handles discrete and stochastic hyperparameters. We call the resulting networks, which update their own hyperparameters online during training, Self-Tuning Networks (STNs).

### 3.1 AN EFFICIENT BEST-RESPONSE APPROXIMATION FOR NEURAL NETWORKS

We propose to approximate the best-response for a given layer's weight matrix $\boldsymbol{W} \in \mathbb{R}^{D_{out} \times D_{in}}$ and bias $\boldsymbol{b} \in \mathbb{R}^{D_{out}}$ as an affine transformation of the hyperparameters $\boldsymbol{\lambda}$[3]:

$$\hat{\boldsymbol{W}}_\phi(\boldsymbol{\lambda}) = \boldsymbol{W}_{\text{elem}} + (\boldsymbol{V}\boldsymbol{\lambda}) \odot_{\text{row}} \boldsymbol{W}_{\text{hyper}}, \quad \hat{\boldsymbol{b}}_\phi(\boldsymbol{\lambda}) = \boldsymbol{b}_{\text{elem}} + (\boldsymbol{C}\boldsymbol{\lambda}) \odot \boldsymbol{b}_{\text{hyper}} \tag{10}$$

Here, $\odot$ indicates elementwise multiplication and $\odot_{\text{row}}$ indicates row-wise rescaling. This architecture computes the usual elementary weight/bias, plus an additional weight/bias which has been scaled by a linear transformation of the hyperparameters. Alternatively, it can be interpreted as directly operating on the pre-activations of the layer, adding a correction to the usual pre-activation to account for the hyperparameters:

$$\hat{\boldsymbol{W}}_\phi(\boldsymbol{\lambda})\boldsymbol{x} + \hat{\boldsymbol{b}}_\phi(\boldsymbol{\lambda}) = [\boldsymbol{W}_{\text{elem}}\boldsymbol{x} + \boldsymbol{b}_{\text{elem}}] + [(\boldsymbol{V}\boldsymbol{\lambda}) \odot (\boldsymbol{W}_{\text{hyper}}\boldsymbol{x}) + (\boldsymbol{C}\boldsymbol{\lambda}) \odot \boldsymbol{b}_{\text{hyper}}] \tag{11}$$

This best-response architecture is tractable to compute and memory-efficient: it requires $D_{out}(2D_{in} + n)$ parameters to represent $\hat{\boldsymbol{W}}_\phi$ and $D_{out}(2 + n)$ parameters to represent $\hat{\boldsymbol{b}}_\phi$, where $n$ is the number of hyperparameters. Furthermore, it enables parallelism: since the predictions can be computed by transforming the pre-activations (Equation 11), the hyperparameters for different examples in a batch can be perturbed independently, improving sample efficiency. In practice, the approximation can be implemented by simply replacing existing modules in deep learning libraries with "hyper" counterparts which accept an additional vector of hyperparameters as input[4].

### 3.2 EXACT BEST-RESPONSE FOR TWO-LAYER LINEAR NETWORKS

Given that the best-response function is a mapping from $\mathbb{R}^n$ to the high-dimensional weight space $\mathbb{R}^m$, why should we expect to be able to represent it compactly? And why in particular would equation 10 be a reasonable approximation? In this section, we exhibit a model whose best-response function can be represented exactly using a minor variant of equation 10: a linear network with Jacobian norm regularization. In particular, the best-response takes the form of a network whose hidden units are modulated conditionally on the hyperparameters.

Consider using a 2-layer linear network with weights $\mathbf{w} = (\boldsymbol{Q}, \boldsymbol{s}) \in \mathbb{R}^{D \times D} \times \mathbb{R}^D$ to predict targets $t \in \mathbb{R}$ from inputs $\boldsymbol{x} \in \mathbb{R}^D$:

---

[3]We describe modifications for convolutional filters in Appendix C.

[4]We illustrate how this is done for the PyTorch library (Paszke et al., 2017) in Appendix G.

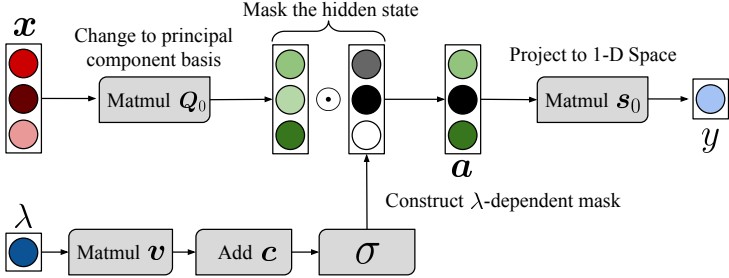

Figure 1: Best-response architecture for an $L_2$-Jacobian regularized two-layer linear network.

$$a(\boldsymbol{x}; \mathbf{w}) = \boldsymbol{Q}\boldsymbol{x}, \quad y(\boldsymbol{x}; \mathbf{w}) = \boldsymbol{s}^\top a(\boldsymbol{x}; \mathbf{w}) \tag{12}$$

Suppose we use a squared-error loss regularized with an $L_2$ penalty on the Jacobian $\partial y/\partial \boldsymbol{x}$, where the penalty weight $\lambda$ lies in $\mathbb{R}$ and is mapped using $\exp$ to lie $\mathbb{R}_+$ :

$$\mathcal{L}_T(\lambda, \mathbf{w}) = \sum_{(\boldsymbol{x},t)\in\mathcal{D}} (y(\boldsymbol{x}; \mathbf{w}) - t)^2 + \frac{1}{|\mathcal{D}|} \exp(\lambda) \left\| \frac{\partial y}{\partial \boldsymbol{x}}(\boldsymbol{x}; \mathbf{w}) \right\|^2 \tag{13}$$

**Theorem 2.** *Let $\boldsymbol{w}_0 = (\boldsymbol{Q}_0, \boldsymbol{s}_0)$, where $\boldsymbol{Q}_0$ is the change-of-basis matrix to the principal components of the data matrix and $\boldsymbol{s}_0$ solves the unregularized version of Problem 13 given $\boldsymbol{Q}_0$. Then there exist $\boldsymbol{v}, \boldsymbol{c} \in \mathbb{R}^D$ such that the best-response function[5] $\boldsymbol{w}^*(\lambda) = (\boldsymbol{Q}^*(\lambda), \boldsymbol{s}^*(\lambda))$ is:*

$$\boldsymbol{Q}^*(\lambda) = \sigma(\lambda\boldsymbol{v} + \boldsymbol{c}) \odot_{\mathrm{row}} \boldsymbol{Q}_0, \quad \boldsymbol{s}^*(\lambda) = \boldsymbol{s}_0,$$

*where $\sigma$ is the sigmoid function.*

*Proof.* See Appendix B.2. $\qquad\square$

Observe that $y(\boldsymbol{x}; \mathbf{w}^*(\lambda))$ can be implemented as a regular network with weights $\mathbf{w}_0 = (\boldsymbol{Q}_0, \boldsymbol{s}_0)$ with an additional sigmoidal gating of its hidden units $a(\boldsymbol{x}; \mathbf{w}^*(\lambda))$:

$$a(\boldsymbol{x}; \mathbf{w}^*(\lambda)) = \boldsymbol{Q}^*(\lambda)\boldsymbol{x} = \sigma(\lambda\boldsymbol{v} + \boldsymbol{c}) \odot_{\mathrm{row}} (\boldsymbol{Q}_0\boldsymbol{x}) = \sigma(\lambda\boldsymbol{v} + \boldsymbol{c}) \odot_{\mathrm{row}} a(\boldsymbol{x}; \mathbf{w}_0) \tag{14}$$

This architecture is shown in Figure 1. Inspired by this example, we use a similar gating of the hidden units to approximate the best-response for deep, nonlinear networks.

### 3.3 LINEAR BEST-RESPONSE APPROXIMATIONS

The sigmoidal gating architecture of the preceding section can be further simplified if one only needs to approximate the best-response function for a small range of hyperparameter values. In particular, for a narrow enough hyperparameter distribution, a smooth best-response function can be approximated by an affine function (i.e. its first-order Taylor approximation). Hence, we replace the sigmoidal gating with linear gating, in order that the weights be affine in the hyperparameters. The following theorem shows that, for quadratic lower-level objectives, using an affine approximation to the best-response function and minimizing $\mathbb{E}_{\boldsymbol{\epsilon}\sim p(\boldsymbol{\epsilon}|\boldsymbol{\sigma})} [f(\boldsymbol{\lambda} + \boldsymbol{\epsilon}, \hat{\mathbf{w}}_\phi(\boldsymbol{\lambda} + \boldsymbol{\epsilon}))]$ yields the correct best-response Jacobian, thus ensuring gradient descent on the approximate objective $F(\boldsymbol{\lambda}, \hat{\mathbf{w}}_\phi(\boldsymbol{\lambda}))$ converges to a local optimum:

**Theorem 3.** *Suppose $f$ is quadratic with $\partial^2 f/\partial w^2 \succ \mathbf{0}$, $p(\boldsymbol{\epsilon}|\sigma)$ is Gaussian with mean $\mathbf{0}$ and variance $\sigma^2\boldsymbol{I}$, and $\hat{\boldsymbol{w}}_\phi$ is affine. Fix $\boldsymbol{\lambda}_0 \in \mathbb{R}^n$ and let $\boldsymbol{\phi}^* = \arg\min_\phi \mathbb{E}_{\boldsymbol{\epsilon}\sim p(\boldsymbol{\epsilon}|\boldsymbol{\sigma})} [f(\boldsymbol{\lambda} + \boldsymbol{\epsilon}, \hat{\boldsymbol{w}}_\phi(\boldsymbol{\lambda} + \boldsymbol{\epsilon}))]$. Then we have $\partial \hat{\boldsymbol{w}}_\phi/\partial \boldsymbol{\lambda}(\boldsymbol{\lambda}_0) = \partial w^*/\partial \boldsymbol{\lambda}(\boldsymbol{\lambda}_0)$.*

*Proof.* See Appendix B.3. $\qquad\square$

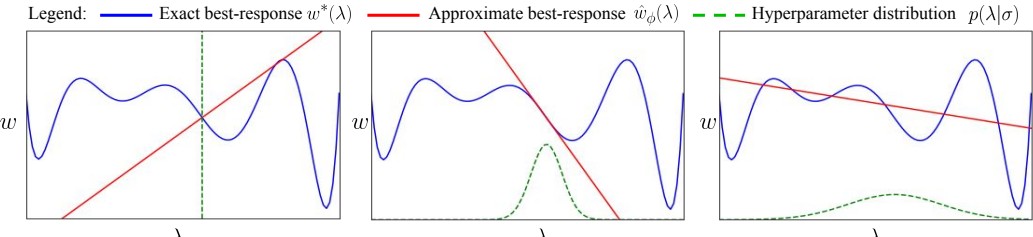

Figure 2: **The effect of the sampled neighborhood. Left:** If the sampled neighborhood is too small (e.g., a point mass) the approximation learned will only match the exact best-response at the current hyperparameter, with no guarantee that its gradient matches that of the best-response. **Middle:** If the sampled neighborhood is not too small or too wide, the gradient of the approximation will match that of the best-response. **Right:** If the sampled neighborhood is too wide, the approximation will be insufficiently flexible to model the best-response, and again the gradients will not match.

## 3.4 Adapting the Hyperparameter Distribution

The entries of $\boldsymbol{\sigma}$ control the scale of the hyperparameter distribution on which $\phi$ is trained. If the entries are too large, then $\hat{\mathbf{w}}_\phi$ will not be flexible enough to capture the best-response over the samples. However, the entries must remain large enough to force $\hat{\mathbf{w}}_\phi$ to capture the shape locally around the current hyperparameter values. We illustrate this in Figure 2. As the smoothness of the loss landscape changes during training, it may be beneficial to vary $\boldsymbol{\sigma}$.

To address these issues, we propose adjusting $\boldsymbol{\sigma}$ during training based on the sensitivity of the upper-level objective to the sampled hyperparameters. We include an entropy term weighted by $\tau \in \mathbb{R}_+$ which acts to enlarge the entries of $\boldsymbol{\sigma}$. The resulting objective is:

$$\mathbb{E}_{\boldsymbol{\epsilon} \sim p(\boldsymbol{\epsilon}|\boldsymbol{\sigma})}[F(\boldsymbol{\lambda} + \boldsymbol{\epsilon}, \hat{\mathbf{w}}_\phi(\boldsymbol{\lambda} + \boldsymbol{\epsilon}))] - \tau \mathbb{H}[p(\boldsymbol{\epsilon}|\boldsymbol{\sigma})] \tag{15}$$

This is similar to a variational inference objective, where the first term is analogous to the negative log-likelihood, but $\tau \neq 1$. As $\tau$ ranges from 0 to 1, our objective interpolates between variational optimization (Staines & Barber, 2012) and variational inference, as noted by Khan et al. (2018). Similar objectives have been used in the variational inference literature for better training (Blundell et al., 2015) and representation learning (Higgins et al., 2017).

Minimizing the first term on its own eventually moves all probability mass towards an optimum $\boldsymbol{\lambda}^*$, resulting in $\boldsymbol{\sigma} = \mathbf{0}$ if $\boldsymbol{\lambda}^*$ is an isolated local minimum. This compels $\boldsymbol{\sigma}$ to balance between shrinking to decrease the first term while remaining sufficiently large to avoid a heavy entropy penalty. When benchmarking our algorithm's performance, we evaluate $F(\boldsymbol{\lambda}, \hat{\mathbf{w}}_\phi(\boldsymbol{\lambda}))$ at the deterministic current hyperparameter $\boldsymbol{\lambda}_0$. (This is a common practice when using stochastic operations during training, such as batch normalization or dropout.)

## 3.5 Training Algorithm

We now describe the complete STN training algorithm and discuss how it can tune hyperparameters that other gradient-based algorithms cannot, such as discrete or stochastic hyperparameters. We use an unconstrained parametrization $\boldsymbol{\lambda} \in \mathbb{R}^n$ of the hyperparameters. Let $r$ denote the element-wise function which maps $\boldsymbol{\lambda}$ to the appropriate constrained space, which will involve a non-differentiable discretization for discrete hyperparameters.

Let $\mathcal{L}_T$ and $\mathcal{L}_V$ denote training and validation losses which are (possibly stochastic, e.g., if using dropout) functions of the hyperparameters and parameters. Define functions $f, F$ by $f(\boldsymbol{\lambda}, \mathbf{w}) = \mathcal{L}_T(r(\boldsymbol{\lambda}), \mathbf{w})$ and $F(\boldsymbol{\lambda}, \mathbf{w}) = \mathcal{L}_V(r(\boldsymbol{\lambda}), \mathbf{w})$. STNs are trained by a gradient descent scheme which alternates between updating $\phi$ for $T_{train}$ steps to minimize $\mathbb{E}_{\boldsymbol{\epsilon} \sim p(\boldsymbol{\epsilon}|\boldsymbol{\sigma})}[f(\boldsymbol{\lambda} + \boldsymbol{\epsilon}, \hat{\mathbf{w}}_\phi(\boldsymbol{\lambda} + \boldsymbol{\epsilon}))]$ (Eq. 9) and updating $\boldsymbol{\lambda}$ and $\boldsymbol{\sigma}$ for $T_{valid}$ steps to minimize $\mathbb{E}_{\boldsymbol{\epsilon} \sim p(\boldsymbol{\epsilon}|\boldsymbol{\sigma})}[F(\boldsymbol{\lambda} + \boldsymbol{\epsilon}, \hat{\mathbf{w}}_\phi(\boldsymbol{\lambda} + \boldsymbol{\epsilon}))] - \tau \mathbb{H}[p(\boldsymbol{\epsilon}|\boldsymbol{\sigma})]$ (Eq. 15). We give our complete algorithm as Algorithm 1 and show how it can be implemented in code in Appendix G. The possible non-differentiability of $r$ due to discrete hyperparameters poses no problem. To estimate the derivative of $\mathbb{E}_{\boldsymbol{\epsilon} \sim p(\boldsymbol{\epsilon}|\boldsymbol{\sigma})}[f(\boldsymbol{\lambda} + \boldsymbol{\epsilon}, \hat{\mathbf{w}}_\phi(\boldsymbol{\lambda} + \boldsymbol{\epsilon}))]$ with respect to $\phi$, we can use the reparametrization trick and compute $\partial f / \partial \mathbf{w}$ and $\partial \hat{\mathbf{w}}_\phi / \partial \phi$, neither of whose computation paths involve the discretization $r$. To differentiate $\mathbb{E}_{\boldsymbol{\epsilon} \sim p(\boldsymbol{\epsilon}|\boldsymbol{\sigma})}[F(\boldsymbol{\lambda} + \boldsymbol{\epsilon}, \hat{\mathbf{w}}_\phi(\boldsymbol{\lambda} + \boldsymbol{\epsilon}))] - \tau \mathbb{H}[p(\boldsymbol{\epsilon}|\boldsymbol{\sigma})]$ with respect to a discrete hyperparameter $\lambda_i$, there are two cases we must consider:

---

[5]This is an abuse of notation since there is not a unique solution to Problem 13 for each $\lambda$ in general.

---

**Algorithm 1** STN Training Algorithm

---

**Initialize:** Best-response approximation parameters $\phi$, hyperparameters $\boldsymbol{\lambda}$, learning rates $\{\alpha_i\}_{i=1}^3$
**while** not converged **do**
    **for** $t = 1, \ldots, T_{train}$ **do**
        $\boldsymbol{\epsilon} \sim p(\boldsymbol{\epsilon}|\boldsymbol{\sigma})$
        $\phi \leftarrow \phi - \alpha_1 \frac{\partial}{\partial \phi} f(\boldsymbol{\lambda} + \boldsymbol{\epsilon}, \hat{\mathbf{w}}_\phi(\boldsymbol{\lambda} + \boldsymbol{\epsilon}))$
    **for** $t = 1, \ldots, T_{valid}$ **do**
        $\boldsymbol{\epsilon} \sim p(\boldsymbol{\epsilon}|\boldsymbol{\sigma})$
        $\boldsymbol{\lambda} \leftarrow \boldsymbol{\lambda} - \alpha_2 \frac{\partial}{\partial \boldsymbol{\lambda}} \left( F(\boldsymbol{\lambda} + \boldsymbol{\epsilon}, \hat{\mathbf{w}}_\phi(\boldsymbol{\lambda} + \boldsymbol{\epsilon})) - \tau \mathbb{H}[p(\boldsymbol{\epsilon}|\boldsymbol{\sigma})] \right)$
        $\boldsymbol{\sigma} \leftarrow \boldsymbol{\sigma} - \alpha_3 \frac{\partial}{\partial \boldsymbol{\sigma}} \left( F(\boldsymbol{\lambda} + \boldsymbol{\epsilon}, \hat{\mathbf{w}}_\phi(\boldsymbol{\lambda} + \boldsymbol{\epsilon})) - \tau \mathbb{H}[p(\boldsymbol{\epsilon}|\boldsymbol{\sigma})] \right)$

---

**Case 1:** For most regularization schemes, $\mathcal{L}_V$ and hence $F$ does not depend on $\lambda_i$ directly and thus the only gradient is through $\hat{\mathbf{w}}_\phi$. Thus, the reparametrization gradient can be used.

**Case 2:** If $\mathcal{L}_V$ relies explicitly on $\lambda_i$, then we can use the REINFORCE gradient estimator (Williams, 1992) to estimate the derivative of the expectation with respect to $\lambda_i$. The number of hidden units in a layer is an example of a hyperparameter that requires this approach since it directly affects the validation loss. We do not show this in Algorithm 1, since we do not tune any hyperparameters which fall into this case.

## 4 EXPERIMENTS

We applied our method to convolutional networks and LSTMs (Hochreiter & Schmidhuber, 1997), yielding self-tuning CNNs (ST-CNNs) and self-tuning LSTMs (ST-LSTMs). We first investigated the behavior of STNs in a simple setting where we tuned a single hyperparameter, and found that STNs discovered *hyperparameter schedules* that outperformed fixed hyperparameter values. Next, we compared the performance of STNs to commonly-used hyperparameter optimization methods on the CIFAR-10 (Krizhevsky & Hinton, 2009) and PTB (Marcus et al., 1993) datasets.

### 4.1 HYPERPARAMETER SCHEDULES

Due to the joint optimization of the hypernetwork weights and hyperparameters, STNs do not use a single, fixed hyperparameter during training. Instead, STNs discover schedules for adapting the hyperparameters online, which can outperform *any* fixed hyperparameter. We examined this behavior in detail on the PTB corpus (Marcus et al., 1993) using an ST-LSTM to tune the output dropout rate applied to the hidden units.

The schedule discovered by an ST-LSTM for output dropout, shown in Figure 3, outperforms the best, fixed output dropout rate (0.68) found by a fine-grained grid search, achieving 82.58 vs 85.83 validation perplexity. We claim that this is a consequence of the schedule, and not of regularizing effects from sampling hyperparameters or the limited capacity of $\hat{\mathbf{w}}_\phi$.

To rule out the possibility that the improved performance is due to stochasticity introduced by sampling hyperparameters during STN training, we trained a standard LSTM while perturbing its dropout rate around the best value found by grid search. We used (1) random Gaussian perturbations, and (2) sinusoid perturbations for a cyclic regularization schedule. STNs outperformed both perturbation methods (Table 1), showing that the improvement is not merely due to hyperparameter stochasticity. Details and plots of each perturbation method are provided in Appendix F.

| Method | Val | Test |
|---|---|---|
| $p = 0.68$, Fixed | 85.83 | 83.19 |
| $p = 0.68$ w/ Gaussian Noise | 85.87 | 82.29 |
| $p = 0.68$ w/ Sinusoid Noise | 85.29 | 82.15 |
| $p = 0.78$ (Final STN Value) | 89.65 | 86.90 |
| **STN** | **82.58** | **79.02** |
| LSTM w/ STN Schedule | 82.87 | 79.93 |

Table 1: Comparing an LSTM trained with fixed and perturbed output dropouts, an STN, and LSTM trained with the STN schedule.

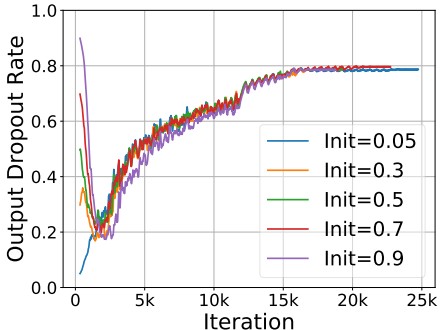

Figure 3: Dropout schedules found by the ST-LSTM for different initial dropout rates.

| | | PTB | | CIFAR-10 | |
|---|---|---|---|---|---|
| **Method** | **Val Perplexity** | **Test Perplexity** | | **Val Loss** | **Test Loss** |
| **Grid Search** | 97.32 | 94.58 | | 0.794 | 0.809 |
| **Random Search** | 84.81 | 81.46 | | 0.921 | 0.752 |
| **Bayesian Optimization** | 72.13 | 69.29 | | 0.636 | 0.651 |
| **STN** | **70.30** | **67.68** | | **0.575** | **0.576** |

Table 2: Final validation and test performance of each method on the PTB word-level language modeling task, and the CIFAR-10 image-classification task.

To determine whether the limited capacity of $\hat{\mathbf{w}}_\phi$ acts as a regularizer, we trained a standard LSTM from scratch using the schedule for output dropout discovered by the ST-LSTM. Using this schedule, the standard LSTM performed nearly as well as the STN, providing evidence that the schedule itself (rather than some other aspect of the STN) was responsible for the improvement over a fixed dropout rate. To further demonstrate the importance of the hyperparameter schedule, we also trained a standard LSTM from scratch using the final dropout value found by the STN (0.78), and found that it did not perform as well as when following the schedule. The final validation and test perplexities of each variant are shown in Table 1.

Next, we show in Figure 3 that the STN discovers the same schedule regardless of the initial hyperparameter values. Because hyperparameters adapt over a shorter timescale than the weights, we find that at any given point in training, the hyperparameter adaptation has already equilibrated. As shown empirically in Appendix F, low regularization is best early in training, while higher regularization is better later on. We found that the STN schedule implements a curriculum by using a low dropout rate early in training, aiding optimization, and then gradually increasing the dropout rate, leading to better generalization.

## 4.2 LANGUAGE MODELING

We evaluated an ST-LSTM on the PTB corpus (Marcus et al., 1993), which is widely used as a benchmark for RNN regularization due to its small size (Gal & Ghahramani, 2016; Merity et al., 2018; Wen et al., 2018). We used a 2-layer LSTM with 650 hidden units per layer and 650-dimensional word embeddings. We tuned 7 hyperparameters: variational dropout rates for the input, hidden state, and output; embedding dropout (that sets rows of the embedding matrix to **0**); Drop-Connect (Wan et al., 2013) on the hidden-to-hidden weight matrix; and coefficients $\alpha$ and $\beta$ that control the strength of activation regularization and temporal activation regularization, respectively. For LSTM tuning, we obtained the best results when using a fixed perturbation scale of 1 for the hyperparameters. Additional details about the experimental setup and the role of these hyperparameters can be found in Appendix D.

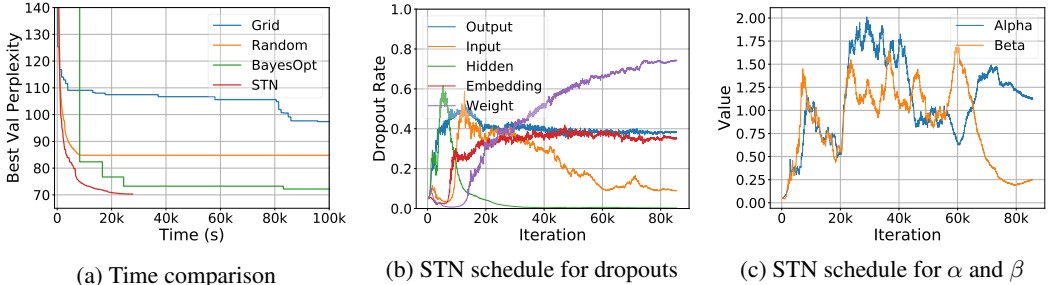

(a) Time comparison   (b) STN schedule for dropouts   (c) STN schedule for $\alpha$ and $\beta$

Figure 4: **(a)** A comparison of the best validation perplexity achieved on PTB over time, by grid search, random search, Bayesian optimization, and STNs. STNs achieve better (lower) validation perplexity in less time than the other methods. **(b)** The hyperparameter schedule found by the STN for each type of dropout. **(c)** The hyperparameter schedule found by the STN for the coefficients of activation regularization and temporal activation regularization.

We compared STNs to grid search, random search, and Bayesian optimization.[6] Figure 4a shows the best validation perplexity achieved by each method over time. STNs outperform other meth-

---

[6]For grid search and random search we used the Ray Tune libraries (https://github.com/ray-project/ray/tree/master/python/ray/tune). For Bayesian optimization, we used Spearmint (https://github.com/HIPS/Spearmint).

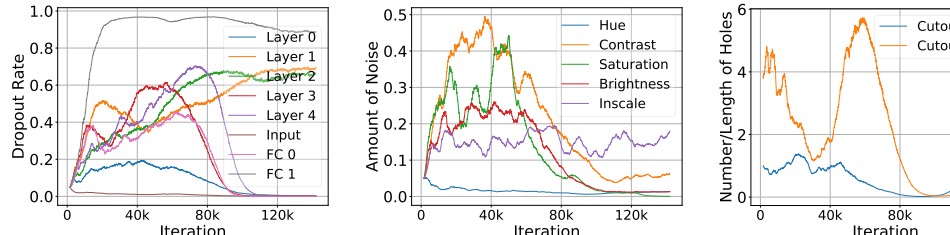

Figure 6: The hyperparameter schedule prescribed by the STN while training for image classification. The dropouts are indexed by the convolutional layer they are applied to. FC dropout is for the fully-connected layers.

ods, achieving lower validation perplexity more quickly. The final validation and test perplexities achieved by each method are shown in Table 2. We show the schedules the STN finds for each hyperparameter in Figures 4b and 4c; we observe that they are nontrivial, with some forms of dropout used to a greater extent at the start of training (including input and hidden dropout), some used throughout training (output dropout), and some that are increased over the course of training (embedding and weight dropout).

## 4.3 IMAGE CLASSIFICATION

We evaluated ST-CNNs on the CIFAR-10 (Krizhevsky & Hinton, 2009) dataset, where it is easy to overfit with high-capacity networks. We used the AlexNet architecture (Krizhevsky et al., 2012), and tuned: (1) continuous hyperparameters controlling per-layer activation dropout, input dropout, and scaling noise applied to the input, (2) discrete data augmentation hyperparameters controlling the length and number of cut-out holes (DeVries & Taylor, 2017), and (3) continuous data augmentation hyperparameters controlling the amount of noise to apply to the hue, saturation, brightness, and contrast of an image. In total, we considered 15 hyperparameters.

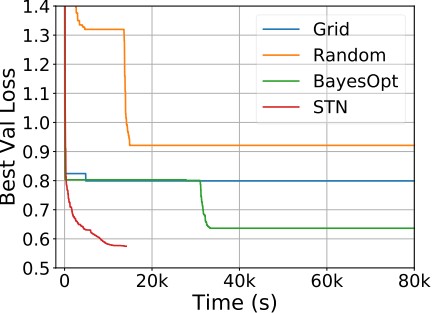

Figure 5: A comparison of the best validation loss achieved on CIFAR-10 over time, by grid search, random search, Bayesian optimization, and STNs. STNs outperform other methods for many computational budgets.

We compared STNs to grid search, random search, and Bayesian optimization. Figure 5 shows the lowest validation loss achieved by each method over time, and Table 2 shows the final validation and test losses for each method. Details of the experimental setup are provided in Appendix E. Again, STNs find better hyperparameter configurations in less time than other methods. The hyperparameter schedules found by the STN are shown in Figure 6.

## 5 RELATED WORK

**Bilevel Optimization.** Colson et al. (2007) provide an overview of bilevel problems, and a comprehensive textbook was written by Bard (2013). When the objectives/constraints are restricted to be linear, quadratic, or convex, a common approach replaces the lower-level problem with its KKT conditions added as constraints for the upper-level problem (Hansen et al., 1992; Vicente et al., 1994). In the unrestricted setting, our work loosely resembles trust-region methods (Colson et al., 2005), which repeatedly approximate the problem locally using a simpler bilevel program. In closely related work, Sinha et al. (2013) used evolutionary techniques to estimate the best-response function iteratively.

**Hypernetworks.** First considered by Schmidhuber (1993; 1992), hypernetworks are functions mapping to the weights of a neural net. Predicting weights in CNNs has been developed in various forms (Denil et al., 2013; Yang et al., 2015). Ha et al. (2016) used hypernetworks to generate weights for modern CNNs and RNNs. Brock et al. (2017) used hypernetworks to globally approximate a best-response for architecture search. Because the architecture is not optimized during training, they require a large hypernetwork, unlike ours which locally approximates the best-response.

**Gradient-Based Hyperparameter Optimization.** There are two main approaches. The first approach approximates $\mathbf{w}^*(\boldsymbol{\lambda}_0)$ using $\mathbf{w}_T(\boldsymbol{\lambda}_0, \mathbf{w}_0)$, the value of $\mathbf{w}$ after $T$ steps of gradient descent on $f$ with respect to $\mathbf{w}$ starting at $(\boldsymbol{\lambda}_0, \mathbf{w}_0)$. The descent steps are differentiated through to approximate $\partial \mathbf{w}^*/\partial \boldsymbol{\lambda}(\boldsymbol{\lambda}_0) \approx \partial \mathbf{w}_T/\partial \boldsymbol{\lambda}(\boldsymbol{\lambda}_0, \mathbf{w}_0)$. This approach was proposed by Domke (2012) and used by Maclaurin et al. (2015), Luketina et al. (2016) and Franceschi et al. (2018). The second approach uses the Implicit Function Theorem to derive $\partial \mathbf{w}^*/\partial \boldsymbol{\lambda}(\boldsymbol{\lambda}_0)$ under certain conditions. This was first developed for hyperparameter optimization in neural networks (Larsen et al., 1996) and developed further by Pedregosa (2016). Similar approaches have been used for hyperparameter optimization in log-linear models (Foo et al., 2008), kernel selection (Chapelle et al., 2002; Seeger, 2007), and image reconstruction (Kunisch & Pock, 2013; Calatroni et al., 2015). Both approaches struggle with certain hyperparameters, since they differentiate gradient descent or the training loss with respect to the hyperparameters. In addition, differentiating gradient descent becomes prohibitively expensive as the number of descent steps increases, while implicitly deriving $\partial \mathbf{w}^*/\partial \boldsymbol{\lambda}$ requires using Hessian-vector products with conjugate gradient solvers to avoid directly computing the Hessian.

**Model-Based Hyperparameter Optimization.** A common model-based approach is Bayesian optimization, which models $p(r|\boldsymbol{\lambda}, \mathcal{D})$, the conditional probability of the performance on some metric $r$ given hyperparameters $\boldsymbol{\lambda}$ and a dataset $\mathcal{D} = \{(\boldsymbol{\lambda}_i, r_i)\}$. We can model $p(r|\boldsymbol{\lambda}, \mathcal{D})$ with various methods (Hutter et al., 2011; Bergstra et al., 2011; Snoek et al., 2012; 2015). $\mathcal{D}$ is constructed iteratively, where the next $\boldsymbol{\lambda}$ to train on is chosen by maximizing an acquisition function $C(\boldsymbol{\lambda}; p(r|\boldsymbol{\lambda}, \mathcal{D}))$ which balances exploration and exploitation. Training each model to completion can be avoided if assumptions are made on learning curve behavior (Swersky et al., 2014; Klein et al., 2017). These approaches require building inductive biases into $p(r|\boldsymbol{\lambda}, \mathcal{D})$ which may not hold in practice, do not take advantage of the network structure when used for hyperparameter optimization, and do not scale well with the number of hyperparameters. However, these approaches have consistency guarantees in the limit, unlike ours.

**Model-Free Hyperparameter Optimization.** Model-free approaches include grid search and random search. Bergstra & Bengio (2012) advocated using random search over grid search. Successive Halving (Jamieson & Talwalkar, 2016) and Hyperband (Li et al., 2017) extend random search by adaptively allocating resources to promising configurations using multi-armed bandit techniques. These methods ignore structure in the problem, unlike ours which uses rich gradient information. However, it is trivial to parallelize model-free methods over computing resources and they tend to perform well in practice.

**Hyperparameter Scheduling.** Population Based Training (PBT) (Jaderberg et al., 2017) considers schedules for hyperparameters. In PBT, a population of networks is trained in parallel. The performance of each network is evaluated periodically, and the weights of under-performing networks are replaced by the weights of better-performing ones; the hyperparameters of the better network are also copied and randomly perturbed for training the new network clone. In this way, a single model can experience different hyperparameter settings over the course of training, implementing a schedule. STNs replace the population of networks by a single best-response approximation and use gradients to tune hyperparameters during a single training run.

## 6 CONCLUSION

We introduced Self-Tuning Networks (STNs), which efficiently approximate the best-response of parameters to hyperparameters by scaling and shifting their hidden units. This allowed us to use gradient-based optimization to tune various regularization hyperparameters, including discrete hyperparameters. We showed that STNs discover hyperparameter schedules that can outperform fixed hyperparameters. We validated the approach on large-scale problems and showed that STNs achieve better generalization performance than competing approaches, in less time. We believe STNs offer a compelling path towards large-scale, automated hyperparameter tuning for neural networks.

ACKNOWLEDGMENTS

We thank Matt Johnson for helpful discussions and advice. MM is supported by an NSERC CGS-M award, and PV is supported by an NSERC PGS-D award. RG acknowledges support from the CIFAR Canadian AI Chairs program.

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

# A TABLE OF NOTATION

Table 3: Table of Notation

| | |
|---|---|
| $\boldsymbol{\lambda}, \mathbf{w}$ | Hyperparameters and parameters |
| $\boldsymbol{\lambda}_0, \mathbf{w}_0$ | Current, fixed hyperparameters and parameters |
| $n, m$ | Hyperparameter and elementary parameter dimension |
| $f(\boldsymbol{\lambda}, \mathbf{w}), F(\boldsymbol{\lambda}, \mathbf{w})$ | Lower-level & upper-level objective |
| $r$ | Function mapping unconstrained hyperparameters to the appropriate restricted space |
| $\mathcal{L}_T(\boldsymbol{\lambda}, \mathbf{w}), \mathcal{L}_V(\boldsymbol{\lambda}, \mathbf{w})$ | Training loss & validation loss - $(\mathcal{L}_T(r(\boldsymbol{\lambda}), \mathbf{w}), \mathcal{L}_V(r(\boldsymbol{\lambda}), \mathbf{w})) = (f(\boldsymbol{\lambda}, \mathbf{w}), F(\boldsymbol{\lambda}, \mathbf{w}))$ |
| $\mathbf{w}^*(\boldsymbol{\lambda})$ | Best-response of the parameters to the hyperparameters |
| $F^*(\boldsymbol{\lambda})$ | Single-level objective from best-response, equals $F(\boldsymbol{\lambda}, \mathbf{w}^*(\boldsymbol{\lambda}))$ |
| $\boldsymbol{\lambda}^*$ | Optimal hyperparameters |
| $\hat{\mathbf{w}}_\phi(\boldsymbol{\lambda})$ | Parametric approximation to the best-response function |
| $\phi$ | Approximate best-response parameters |
| $\boldsymbol{\sigma}$ | Scale of the hyperparameter noise distribution |
| $\sigma$ | The sigmoid function |
| $\boldsymbol{\epsilon}$ | Sampled perturbation noise, to be added to hyperparameters |
| $p(\boldsymbol{\epsilon}\|\boldsymbol{\sigma}), p(\boldsymbol{\lambda}\|\boldsymbol{\sigma})$ | The noise distribution and induced hyperparameter distribution |
| $\alpha$ | A learning rate |
| $T_{train}, T_{valid}$ | Number of training steps on the training and validation data |
| $\boldsymbol{x}, t$ | An input datapoint and its associated target |
| $\mathcal{D}$ | A data set consisting of tuples of inputs and targets |
| $D$ | The dimensionality of input data |
| $y(\boldsymbol{x}, \mathbf{w})$ | Prediction function for input data $\boldsymbol{x}$ and elementary parameters $\mathbf{w}$ |
| $\odot_{\mathrm{row}}$ | Row-wise rescaling - *not elementwise multiplication* |
| $\boldsymbol{Q}, \boldsymbol{s}$ | First and second layer weights of the linear network in Problem 13 |
| $\boldsymbol{Q}_0, \boldsymbol{s}_0$ | The basis change matrix and solution to the unregularized Problem 13 |
| $\boldsymbol{Q}^*(\boldsymbol{\lambda}), \boldsymbol{s}^*(\boldsymbol{\lambda})$ | The best response weights of the linear network in Problem 13 |
| $\boldsymbol{a}(\boldsymbol{x}; \mathbf{w})$ | Activations of hidden units in the linear network of Problem 13 |
| $\boldsymbol{W}, \boldsymbol{b}$ | A layer's weight matrix and bias |
| $D_{out}, D_{in}$ | A layer's input dimensionality and output dimensionality |
| $\frac{\partial \mathcal{L}_T(\boldsymbol{\lambda}, \mathbf{w})}{\partial \boldsymbol{\lambda}}$ | The (validation loss) direct (hyperparameter) gradient |
| $\frac{\partial \mathbf{w}^*(\boldsymbol{\lambda})}{\partial \boldsymbol{\lambda}}$ | The (elementary parameter) response gradient |
| $\frac{\partial \mathcal{L}_T(\boldsymbol{\lambda}, \mathbf{w}^*(\boldsymbol{\lambda}))}{\partial \mathbf{w}^*(\boldsymbol{\lambda})} \frac{\partial \mathbf{w}^*(\boldsymbol{\lambda})}{\partial \boldsymbol{\lambda}}$ | The (validation loss) response gradient |
| $\frac{d\mathcal{L}_T(\boldsymbol{\lambda}, \mathbf{w})}{d\boldsymbol{\lambda}}$ | The hyperparameter gradient: a sum of the validation losses direct and response gradients |

# B PROOFS

## B.1 LEMMA 1

Because $\mathbf{w}_0$ solves Problem 4b given $\boldsymbol{\lambda}_0$, by the first-order optimality condition we must have:

$$\frac{\partial f}{\partial \mathbf{w}}(\boldsymbol{\lambda}_0, \mathbf{w}_0) = 0 \tag{16}$$

The Jacobian of $\partial f / \partial \mathbf{w}$ decomposes as a block matrix with sub-blocks given by:

$$\left[ \begin{array}{c|c} \frac{\partial^2 f}{\partial \boldsymbol{\lambda} \partial \mathbf{w}} & \frac{\partial^2 f}{\partial \mathbf{w}^2} \end{array} \right] \tag{17}$$

We know that $f$ is $\mathcal{C}^2$ in some neighborhood of $(\boldsymbol{\lambda}_0, \mathbf{w}_0)$, so $\partial f / \partial \mathbf{w}$ is continuously differentiable in this neighborhood. By assumption, the Hessian $\partial^2 f / \partial \mathbf{w}^2$ is positive definite and hence invertible at $(\boldsymbol{\lambda}_0, \mathbf{w}_0)$. By the Implicit Function Theorem, there exists a neighborhood $V$ of $\boldsymbol{\lambda}_0$ and a unique continuously differentiable function $\mathbf{w}^* : V \to \mathbb{R}^m$ such that $\partial f / \partial \mathbf{w}(\boldsymbol{\lambda}, \mathbf{w}^*(\boldsymbol{\lambda})) = 0$ for $\boldsymbol{\lambda} \in V$ and $\mathbf{w}^*(\boldsymbol{\lambda}_0) = \mathbf{w}_0$.

Furthermore, by continuity we know that there is a neighborhood $W_1 \times W_2$ of $(\boldsymbol{\lambda}_0, \mathbf{w}_0)$ such that $\partial^2 f / \partial \mathbf{w}^2$ is positive definite on this neighborhood. Setting $U = V \cap W_1 \cap (\mathbf{w}^*)^{-1}(W_2)$, we can conclude that $\partial^2 f / \partial \mathbf{w}^2(\boldsymbol{\lambda}, \mathbf{w}^*(\boldsymbol{\lambda})) \succ 0$ for all $\boldsymbol{\lambda} \in U$. Combining this with $\partial f / \partial \mathbf{w}(\boldsymbol{\lambda}, \mathbf{w}^*(\boldsymbol{\lambda})) = 0$ and using second-order sufficient optimality conditions, we conclude that $\mathbf{w}^*(\boldsymbol{\lambda})$ is the unique solution to Problem 4b for all $\boldsymbol{\lambda} \in U$.

## B.2 LEMMA 2

This discussion mostly follows from Hastie et al. (2001). We let $\boldsymbol{X} \in \mathbb{R}^{N \times D}$ denote the data matrix where $N$ is the number of training examples and $D$ is the dimensionality of the data. We let $\boldsymbol{t} \in \mathbb{R}^N$ denote the associated targets. We can write the SVD decomposition of $\boldsymbol{X}$ as:

$$\boldsymbol{X} = \boldsymbol{U} \boldsymbol{D} \boldsymbol{V}^\top \tag{18}$$

where $\boldsymbol{U}$ and $\boldsymbol{V}$ are $N \times D$ and $D \times D$ orthogonal matrices and $\boldsymbol{D}$ is a diagonal matrix with entries $d_1 \geq d_2 \geq \cdots \geq d_D > 0$. We next simplify the function $y(\boldsymbol{x}; \mathbf{w})$ by setting $\boldsymbol{u} = \boldsymbol{s}^\top \boldsymbol{Q}$, so that $y(\boldsymbol{x}; \mathbf{w}) = \boldsymbol{s}^\top \boldsymbol{Q} \boldsymbol{x} = \boldsymbol{u}^\top \boldsymbol{x}$. We see that the Jacobian $\partial y / \partial \boldsymbol{x} \equiv \boldsymbol{u}$ is constant, and Problem 13 simplifies to standard $L_2$-regularized least-squares linear regression with the following loss function:

$$\sum_{(\boldsymbol{x},t) \in \mathcal{D}} (\boldsymbol{u}^\top \boldsymbol{x} - t)^2 + \frac{1}{|\mathcal{D}|} \exp(\lambda) \|\boldsymbol{u}\|^2 \tag{19}$$

It is well-known (see Hastie et al. (2001), Chapter 3) that the optimal solution $\boldsymbol{u}^*(\lambda)$ minimizing Equation 19 is given by:

$$\boldsymbol{u}^*(\lambda) = (\boldsymbol{X}^\top \boldsymbol{X} + \exp(\lambda)\boldsymbol{I})^{-1} \boldsymbol{X}^\top \boldsymbol{t} = \boldsymbol{V}(\boldsymbol{D}^2 + \exp(\lambda)\boldsymbol{I})^{-1} \boldsymbol{D} \boldsymbol{U}^\top \boldsymbol{t} \tag{20}$$

Furthermore, the optimal solution $\boldsymbol{u}^*$ to the unregularized version of Problem 19 is given by:

$$\boldsymbol{u}^* = \boldsymbol{V} \boldsymbol{D}^{-1} \boldsymbol{U}^\top \boldsymbol{t} \tag{21}$$

Recall that we defined $\boldsymbol{Q}_0 = \boldsymbol{V}^\top$, i.e., the change-of-basis matrix from the standard basis to the principal components of the data matrix, and we defined $\boldsymbol{s}_0$ to solve the unregularized regression problem given $\boldsymbol{Q}_0$. Thus, we require that $\boldsymbol{Q}_0^\top \boldsymbol{s}_0 = \boldsymbol{u}^*$ which implies $\boldsymbol{s}_0 = \boldsymbol{D}^{-1} \boldsymbol{U}^\top \boldsymbol{t}$.

There are not unique solutions to Problem 13, so we take any functions $\boldsymbol{Q}(\lambda), \boldsymbol{s}(\lambda)$ which satisfy $\boldsymbol{Q}(\lambda)^\top \boldsymbol{s}(\lambda) = \boldsymbol{v}^*(\lambda)$ as "best-response functions". We will show that our chosen functions $\boldsymbol{Q}^*(\lambda) = \sigma(\lambda \boldsymbol{v} + \boldsymbol{c}) \odot_{\text{row}} \boldsymbol{Q}_0$ and $\boldsymbol{s}^*(\lambda) = \boldsymbol{s}_0$, where $\boldsymbol{v} = -\mathbf{1}$ and $c_i = 2 \log(d_i)$ for $i = 1, \ldots, D$, meet this criteria. We start by noticing that for any $d \in \mathbb{R}_+$, we have:

$$\sigma(-\lambda + 2\log(d)) = \frac{1}{1 + \exp(\lambda - 2\log(d))} = \frac{1}{1 + d^{-2}\exp(\lambda)} = \frac{d^2}{d^2 + \exp(\lambda)} \tag{22}$$

It follows that:

$$\boldsymbol{Q}^*(\lambda)^\top \boldsymbol{s}^*(\lambda) = [\sigma(\lambda \boldsymbol{v} + \boldsymbol{c}) \odot_{\mathrm{row}} \boldsymbol{Q}_0]^\top \boldsymbol{s}_0 \tag{23}$$

$$= \left[ \mathrm{diag}\begin{pmatrix} \sigma(-\lambda + 2\log(d_1)) \\ \vdots \\ \sigma(-\lambda + 2\log(d_D)) \end{pmatrix} \boldsymbol{Q}_0 \right]^\top \boldsymbol{s}_0 \tag{24}$$

$$= \boldsymbol{Q}_0^\top \left[ \mathrm{diag}\begin{pmatrix} \frac{d_1^2}{d_1^2 + \exp(\lambda)} \\ \vdots \\ \frac{d_D^2}{d_D^2 + \exp(\lambda)} \end{pmatrix} \right] \boldsymbol{s}_0 \tag{25}$$

$$= \boldsymbol{V} \left[ \mathrm{diag}\begin{pmatrix} \frac{d_1^2}{d_1^2 + \exp(\lambda)} \\ \vdots \\ \frac{d_D^2}{d_D^2 + \exp(\lambda)} \end{pmatrix} \right] \boldsymbol{D}^{-1} \boldsymbol{U}^\top \boldsymbol{t} \tag{26}$$

$$= \boldsymbol{V} \left[ \mathrm{diag}\begin{pmatrix} \frac{d_1}{d_1^2 + \exp(\lambda)} \\ \vdots \\ \frac{d_D}{d_D^2 + \exp(\lambda)} \end{pmatrix} \right] \boldsymbol{U}^\top \boldsymbol{t} \tag{27}$$

$$= \boldsymbol{V} \left[ (\boldsymbol{D}^2 + \exp(\lambda)\boldsymbol{I})^{-1}\boldsymbol{D} \right] \boldsymbol{U}^\top \boldsymbol{t} \tag{28}$$

$$= \boldsymbol{v}^*(\lambda) \tag{29}$$

## B.3 THEOREM 3

By assumption $f$ is quadratic, so there exist $\boldsymbol{A} \in \mathbb{R}^{n \times n}, \boldsymbol{B} \in \mathbb{R}^{n \times m}, \boldsymbol{C} \in \mathbb{R}^{m \times m}$ and $\boldsymbol{d} \in \mathbb{R}^n, \boldsymbol{e} \in \mathbb{R}^m$ such that:

$$f(\boldsymbol{\lambda}, \mathbf{w}) = \frac{1}{2}\begin{pmatrix} \boldsymbol{\lambda}^\top & \mathbf{w}^\top \end{pmatrix}\begin{pmatrix} \boldsymbol{A} & \boldsymbol{B} \\ \boldsymbol{B}^\top & \boldsymbol{C} \end{pmatrix}\begin{pmatrix} \boldsymbol{\lambda} \\ \mathbf{w} \end{pmatrix} + \boldsymbol{d}^\top \boldsymbol{\lambda} + \boldsymbol{e}^\top \mathbf{w} \tag{30}$$

One can easily compute that:

$$\frac{\partial f}{\partial \mathbf{w}}(\boldsymbol{\lambda}, \mathbf{w}) = \boldsymbol{B}^\top \boldsymbol{\lambda} + \boldsymbol{C}\mathbf{w} + \boldsymbol{e} \tag{31}$$

$$\frac{\partial^2 f}{\partial \mathbf{w}^2}(\boldsymbol{\lambda}, \mathbf{w}) = \boldsymbol{C} \tag{32}$$

Since we assume $\partial^2 f / \partial \mathbf{w}^2 \succ \mathbf{0}$, we must have $\boldsymbol{C} \succ \mathbf{0}$. Setting the derivative equal to $\mathbf{0}$ and using second-order sufficient conditions, we have:

$$\mathbf{w}^*(\boldsymbol{\lambda}) = -\boldsymbol{C}^{-1}(\boldsymbol{e} + \boldsymbol{B}^\top \boldsymbol{\lambda}) \tag{33}$$

Hence, we find:

$$\frac{\partial \mathbf{w}^*}{\partial \boldsymbol{\lambda}}(\boldsymbol{\lambda}) = -\boldsymbol{C}^{-1}\boldsymbol{B}^\top \tag{34}$$

We let $\hat{\mathbf{w}}_\phi(\boldsymbol{\lambda}) = \boldsymbol{U}\boldsymbol{\lambda} + \boldsymbol{b}$, and define $\hat{f}$ to be the function given by:

$$\hat{f}(\boldsymbol{\lambda}, \boldsymbol{U}, \boldsymbol{b}, \boldsymbol{\sigma}) = \mathbb{E}_{\boldsymbol{\epsilon} \sim p(\boldsymbol{\epsilon}|\boldsymbol{\sigma})}\left[ f(\boldsymbol{\lambda} + \boldsymbol{\epsilon}, \boldsymbol{U}(\boldsymbol{\lambda} + \boldsymbol{\epsilon}) + \boldsymbol{b}) \right] \tag{35}$$

Substituting and simplifying:

$$\hat{f}(\boldsymbol{\lambda}_0, \boldsymbol{U}, \boldsymbol{b}, \boldsymbol{\sigma}) = \mathbb{E}_{\boldsymbol{\epsilon} \sim p(\boldsymbol{\epsilon}|\boldsymbol{\sigma})}\left[ \frac{1}{2}(\boldsymbol{\lambda}_0 + \boldsymbol{\epsilon})^\top \boldsymbol{A}(\boldsymbol{\lambda}_0 + \boldsymbol{\epsilon}) + (\boldsymbol{\lambda}_0 + \boldsymbol{\epsilon})^\top \boldsymbol{B}(\boldsymbol{U}(\boldsymbol{\lambda}_0 + \boldsymbol{\epsilon}) + \boldsymbol{b}) \right.$$

$$+ \frac{1}{2}(\boldsymbol{U}(\boldsymbol{\lambda}_0 + \boldsymbol{\epsilon}) + \boldsymbol{b})^\top \boldsymbol{C}(\boldsymbol{U}(\boldsymbol{\lambda}_0 + \boldsymbol{\epsilon}) + \boldsymbol{b})$$

$$\left. + \boldsymbol{d}^\top(\boldsymbol{\lambda}_0 + \boldsymbol{\epsilon}) + \boldsymbol{e}^\top(\boldsymbol{U}(\boldsymbol{\lambda}_0 + \boldsymbol{\epsilon}) + \boldsymbol{b}) \right] \tag{36}$$

Expanding, we find that equation 36 is equal to:

$$\mathbb{E}_{\epsilon \sim p(\epsilon|\sigma)} \left[ \textcircled{1} + \textcircled{2} + \textcircled{3} + \textcircled{4} \right] \tag{37}$$

where we have:

$$\textcircled{1} = \frac{1}{2} \left( \boldsymbol{\lambda}_0^\top \boldsymbol{A} \boldsymbol{\lambda}_0 + 2\boldsymbol{\epsilon}^\top \boldsymbol{A} \boldsymbol{\lambda}_0 + \boldsymbol{\epsilon}^\top \boldsymbol{A} \boldsymbol{\epsilon} \right) \tag{38}$$

$$\textcircled{2} = \boldsymbol{\lambda}_0^\top \boldsymbol{B} \boldsymbol{U} \boldsymbol{\lambda}_0 + \boldsymbol{\lambda}_0^\top \boldsymbol{B} \boldsymbol{U} \boldsymbol{\epsilon} + \boldsymbol{\lambda}_0^\top \boldsymbol{B} \boldsymbol{b} + \boldsymbol{\epsilon}^\top \boldsymbol{B} \boldsymbol{U} \boldsymbol{\lambda}_0 + \boldsymbol{\epsilon}^\top \boldsymbol{B} \boldsymbol{U} \boldsymbol{\epsilon} + \boldsymbol{\epsilon}^\top \boldsymbol{B} \boldsymbol{b} \tag{39}$$

$$\textcircled{3} = \frac{1}{2} (\boldsymbol{\lambda}_0^\top \boldsymbol{U}^\top \boldsymbol{C} \boldsymbol{U} \boldsymbol{\lambda}_0 + \boldsymbol{\lambda}_0 \boldsymbol{U}^\top \boldsymbol{C} \boldsymbol{U} \boldsymbol{\epsilon} + \boldsymbol{\lambda}_0 \boldsymbol{U}^\top \boldsymbol{C} \boldsymbol{b} + \boldsymbol{\epsilon}^\top \boldsymbol{U}^\top \boldsymbol{C} \boldsymbol{U} \boldsymbol{\lambda}_0$$
$$+ \boldsymbol{\epsilon}^\top \boldsymbol{U}^\top \boldsymbol{C} \boldsymbol{U} \boldsymbol{\epsilon} + \boldsymbol{\epsilon}^\top \boldsymbol{U}^\top \boldsymbol{C} \boldsymbol{b} + \boldsymbol{b}^\top \boldsymbol{C} \boldsymbol{U} \boldsymbol{\lambda}_0 + \boldsymbol{b}^\top \boldsymbol{C} \boldsymbol{U} \boldsymbol{\epsilon} + \boldsymbol{b}^\top \boldsymbol{C} \boldsymbol{b}) \tag{40}$$

$$\textcircled{4} = \boldsymbol{d}^\top \boldsymbol{\lambda}_0 + \boldsymbol{d}^\top \boldsymbol{\epsilon} + \boldsymbol{e}^\top \boldsymbol{U} \boldsymbol{\lambda}_0 + \boldsymbol{e}^\top \boldsymbol{U} \boldsymbol{\epsilon} + \boldsymbol{e}^\top \boldsymbol{b} \tag{41}$$

We can simplify these expressions considerably by using linearity of expectation and that $\boldsymbol{\epsilon} \sim p(\boldsymbol{\epsilon}|\sigma)$ has mean $\mathbf{0}$:

$$\mathbb{E}_{\epsilon \sim p(\epsilon|\sigma)} \left[ \textcircled{1} \right] = \frac{1}{2} \boldsymbol{\lambda}_0^\top \boldsymbol{A} \boldsymbol{\lambda}_0 \tag{42}$$

$$\mathbb{E}_{\epsilon \sim p(\epsilon|\sigma)} \left[ \textcircled{2} \right] = \boldsymbol{\lambda}_0^\top \boldsymbol{B} \boldsymbol{U} \boldsymbol{\lambda}_0 + \boldsymbol{\lambda}_0^\top \boldsymbol{B} \boldsymbol{b} + \mathbb{E}_{\epsilon \sim p(\epsilon|\sigma)} \left[ \boldsymbol{\epsilon}^\top \boldsymbol{B} \boldsymbol{U} \boldsymbol{\epsilon} \right] \tag{43}$$

$$\mathbb{E}_{\epsilon \sim p(\epsilon|\sigma)} \left[ \textcircled{3} \right] = \frac{1}{2} (\boldsymbol{\lambda}_0^\top \boldsymbol{U}^\top \boldsymbol{C} \boldsymbol{U} \boldsymbol{\lambda}_0 + \boldsymbol{\lambda}_0 \boldsymbol{U}^\top \boldsymbol{C} \boldsymbol{b} +$$
$$\mathbb{E}_{\epsilon \sim p(\epsilon|\sigma)} \left[ \boldsymbol{\epsilon}^\top \boldsymbol{U}^\top \boldsymbol{C} \boldsymbol{U} \boldsymbol{\epsilon} \right] + \boldsymbol{b}^\top \boldsymbol{C} \boldsymbol{U} \boldsymbol{\lambda}_0 + \boldsymbol{b}^\top \boldsymbol{C} \boldsymbol{b}) \tag{44}$$

$$\mathbb{E}_{\epsilon \sim p(\epsilon|\sigma)} \left[ \textcircled{4} \right] = \boldsymbol{d}^\top \boldsymbol{\lambda}_0 + \boldsymbol{e}^\top \boldsymbol{U} \boldsymbol{\lambda}_0 + \boldsymbol{e}^\top \boldsymbol{b} \tag{45}$$

We can use the cyclic property of the Trace operator, $\mathbb{E}_{\epsilon \sim p(\epsilon|\sigma)}[\boldsymbol{\epsilon}\boldsymbol{\epsilon}^\top] = \sigma^2 \boldsymbol{I}$, and commutability of expectation and a linear operator to simplify the expectations of $\textcircled{2}$ and $\textcircled{3}$:

$$\mathbb{E}_{\epsilon \sim p(\epsilon|\sigma)} \left[ \textcircled{2} \right] = \boldsymbol{\lambda}_0^\top \boldsymbol{B} \boldsymbol{U} \boldsymbol{\lambda}_0 + \boldsymbol{\lambda}_0^\top \boldsymbol{B} \boldsymbol{b} + \text{Tr} \left[ \sigma^2 \boldsymbol{B} \boldsymbol{U} \right] \tag{46}$$

$$\mathbb{E}_{\epsilon \sim p(\epsilon|\sigma)} \left[ \textcircled{3} \right] = \frac{1}{2} (\boldsymbol{\lambda}_0^\top \boldsymbol{U}^\top \boldsymbol{C} \boldsymbol{U} \boldsymbol{\lambda}_0 + \boldsymbol{\lambda}_0 \boldsymbol{U}^\top \boldsymbol{C} \boldsymbol{b} +$$
$$\text{Tr} \left[ \sigma^2 \boldsymbol{U}^\top \boldsymbol{C} \boldsymbol{U} \right] + \boldsymbol{b}^\top \boldsymbol{C} \boldsymbol{U} \boldsymbol{\lambda}_0 + \boldsymbol{b}^\top \boldsymbol{C} \boldsymbol{b}) \tag{47}$$

We can then differentiate $\hat{f}$ by making use of various matrix-derivative equalities (Duchi, 2007) to find:

$$\frac{\partial \hat{f}}{\partial \boldsymbol{b}}(\boldsymbol{\lambda}_0, \boldsymbol{U}, \boldsymbol{b}, \sigma) = \frac{1}{2} \boldsymbol{C}^\top \boldsymbol{U} \boldsymbol{\lambda}_0 + \frac{1}{2} \boldsymbol{C} \boldsymbol{U} \boldsymbol{\lambda}_0 + \boldsymbol{B}^\top \boldsymbol{\lambda}_0 + \boldsymbol{e} + \boldsymbol{C} \boldsymbol{b} \tag{48}$$

$$\frac{\partial \hat{f}}{\partial \boldsymbol{U}}(\boldsymbol{\lambda}_0, \boldsymbol{U}, \boldsymbol{b}, \sigma) = \boldsymbol{B}^\top \boldsymbol{\lambda}_0 \boldsymbol{\lambda}_0^\top + \sigma^2 \boldsymbol{B}^\top + \boldsymbol{C} \boldsymbol{b} \boldsymbol{\lambda}_0^\top + \boldsymbol{e} \boldsymbol{\lambda}_0^\top + \boldsymbol{C} \boldsymbol{U} \boldsymbol{\lambda}_0 \boldsymbol{\lambda}_0^\top + \sigma^2 \boldsymbol{C} \boldsymbol{U} \tag{49}$$

Setting the derivative $\partial \hat{f}/\partial \boldsymbol{b}(\boldsymbol{\lambda}_0, \boldsymbol{U}, \boldsymbol{b}, \sigma)$ equal to $\mathbf{0}$, we have:

$$\boldsymbol{b} = -\boldsymbol{C}^{-1}(\boldsymbol{C}^\top \boldsymbol{U} \boldsymbol{\lambda}_0 + \boldsymbol{B}^\top \boldsymbol{\lambda}_0 + \boldsymbol{e}) \tag{50}$$

Setting the derivative for $\partial \hat{f}/\partial \boldsymbol{U}(\boldsymbol{\lambda}_0, \boldsymbol{U}, \boldsymbol{b}, \sigma)$ equal to $\mathbf{0}$, we have:

$$\boldsymbol{C} \boldsymbol{U}(\boldsymbol{\lambda}_0 \boldsymbol{\lambda}_0^\top + \sigma^2 \boldsymbol{C}) = -\boldsymbol{B}^\top \boldsymbol{\lambda}_0 \boldsymbol{\lambda}_0^\top - \sigma^2 \boldsymbol{B}^\top - \boldsymbol{C} \boldsymbol{b} \boldsymbol{\lambda}_0^\top - \boldsymbol{e} \boldsymbol{\lambda}_0^\top \tag{51}$$

Substituting the expression for $\boldsymbol{b}$ given by equation 50 into equation 51 and simplifying gives:

$$\boldsymbol{C} \boldsymbol{U}(\boldsymbol{\lambda}_0 \boldsymbol{\lambda}_0^\top + \sigma^2 \boldsymbol{I}) = -\sigma^2 \boldsymbol{B}^\top + \boldsymbol{C} \boldsymbol{U} \boldsymbol{\lambda}_0 \boldsymbol{\lambda}_0^\top \tag{52}$$

$$\implies \sigma^2 \boldsymbol{C} \boldsymbol{U} = -\sigma^2 \boldsymbol{B}^\top \tag{53}$$

$$\implies \boldsymbol{U} = -\boldsymbol{C}^{-1} \boldsymbol{B}^\top \tag{54}$$

This is exactly the best-response Jacobian $\partial \mathbf{w}^*/\partial \lambda(\boldsymbol{\lambda})$ as given by Equation 34. Substituting $\boldsymbol{U} = \boldsymbol{C}^{-1} \boldsymbol{B}$ into the equation 50 gives:

$$\boldsymbol{b} = \boldsymbol{C}^{-1} \boldsymbol{B}^\top \boldsymbol{\lambda}_0 - \boldsymbol{C}^{-1} \boldsymbol{B}^\top \boldsymbol{\lambda}_0 - \boldsymbol{C}^{-1} \boldsymbol{e} \tag{55}$$

This is $\mathbf{w}^*(\boldsymbol{\lambda}_0) - \partial \mathbf{w}^*/\partial \lambda(\boldsymbol{\lambda}_0)$, thus the approximate best-response is exactly the first-order Taylor series of $\mathbf{w}^*$ about $\boldsymbol{\lambda}_0$.

### B.4 BEST-RESPONSE GRADIENT LEMMA

**Lemma 4.** *Under the same conditions as Lemma 1 and using the same notation, for all $\boldsymbol{\lambda} \in U$, we have that:*

$$\frac{\partial \boldsymbol{w}^*}{\partial \boldsymbol{\lambda}}(\boldsymbol{\lambda}) = -\left[\frac{\partial^2 f}{\partial \boldsymbol{w}^2}(\boldsymbol{\lambda}, \boldsymbol{w}^*(\boldsymbol{\lambda}))\right]^{-1} \frac{\partial^2 f}{\partial \boldsymbol{\lambda} \partial \boldsymbol{w}}(\boldsymbol{\lambda}, \boldsymbol{w}^*(\boldsymbol{\lambda})) \tag{56}$$

*Proof.* Define $\iota^* : U \to \mathbb{R}^n \times \mathbb{R}^m$ by $\iota^*(\boldsymbol{\lambda}) = (\boldsymbol{\lambda}, \mathbf{w}^*(\boldsymbol{\lambda}))$. By first-order optimality conditions, we know that:

$$\left(\frac{\partial f}{\partial \mathbf{w}} \circ \iota^*\right)(\boldsymbol{\lambda}) = 0 \quad \forall \boldsymbol{\lambda} \in U \tag{57}$$

Hence, for all $\boldsymbol{\lambda} \in U$:

$$0 = \frac{\partial}{\partial \boldsymbol{\lambda}}\left(\frac{\partial f}{\partial \mathbf{w}} \circ \iota^*\right)(\boldsymbol{\lambda}) \tag{58}$$

$$= \frac{\partial^2 f}{\partial \mathbf{w}^2}(\iota^*(\boldsymbol{\lambda}))\frac{\partial \mathbf{w}^*}{\partial \boldsymbol{\lambda}}(\boldsymbol{\lambda}) + \frac{\partial^2 f}{\partial \boldsymbol{\lambda} \partial \mathbf{w}}(\iota^*(\boldsymbol{\lambda})) \tag{59}$$

$$= \frac{\partial^2 f}{\partial \mathbf{w}^2}(\boldsymbol{\lambda}, \mathbf{w}^*(\boldsymbol{\lambda}))\frac{\partial \mathbf{w}^*}{\partial \boldsymbol{\lambda}}(\boldsymbol{\lambda}) + \frac{\partial^2 f}{\partial \boldsymbol{\lambda} \partial \mathbf{w}}(\boldsymbol{\lambda}, \mathbf{w}^*(\lambda)) \tag{60}$$

Rearranging gives Equation 56. $\qquad\square$

## C  BEST-RESPONSE APPROXIMATIONS FOR CONVOLUTIONAL FILTERS

We let $L$ denote the number of layers, $C_l$ the number of channels in layer $l$'s feature map, and $K_l$ the size of the kernel in layer $l$. We let $\boldsymbol{W}^{l,c} \in \mathbb{R}^{C_{l-1} \times K_l \times K_l}$ and $\boldsymbol{b}^{l,c} \in \mathbb{R}$ denote the weight and bias respectively of the $c^{\text{th}}$ convolution kernel in layer $l$ (so $c \in \{1, \ldots, C_l\}$). For $\boldsymbol{u}^{l,c}, \boldsymbol{a}^{l,c} \in \mathbb{R}^n$, we define best-response approximations $\hat{\boldsymbol{W}}_{\boldsymbol{\phi}}^{l,c}$ and $\hat{\boldsymbol{b}}_{\boldsymbol{\phi}}^{l,c}$ by:

$$\hat{\boldsymbol{W}}_{\boldsymbol{\phi}}^{l,c}(\boldsymbol{\lambda}) = (\boldsymbol{\lambda}^\top \boldsymbol{u}^{l,c}) \odot \boldsymbol{W}_{\text{hyper}}^{l,c} + \boldsymbol{W}_{\text{elem}}^{l,c} \tag{61}$$

$$\hat{\boldsymbol{b}}_{\boldsymbol{\phi}}^{l,c}(\boldsymbol{\lambda}) = (\boldsymbol{\lambda}^\top \boldsymbol{a}^{l,c}) \odot \boldsymbol{b}_{\text{hyper}}^{l,c} + \boldsymbol{b}_{\text{elem}}^{l,c} \tag{62}$$

Thus, the best-response parameters used for modeling $\boldsymbol{W}^{l,c}$, $\boldsymbol{b}^l$ are $\{\boldsymbol{u}^{l,c}, \boldsymbol{a}^{l,c}, \boldsymbol{W}_{\text{hyper}}^{l,c}, \boldsymbol{W}_{\text{elem}}^{l,c}, \boldsymbol{b}_{\text{hyper}}^{l,c}, \boldsymbol{b}_{\text{elem}}^{l,c}\}$. We can compute the number of parameters used as $2n + 2(|\boldsymbol{W}^{l,c}| + |\boldsymbol{b}^{l,c}|)$. Summing over channels $c$, we find the total number of parameters is $2nC_l + 2p$, where $p$ is the total number of parameters in the normal CNN layer. Hence, we use twice the number of parameters in a normal CNN, plus an overhead that depends on the number of hyperparameters.

For an implementation in code, see Appendix G.

## D  LANGUAGE MODELING EXPERIMENT DETAILS

Here we present additional details on the setup of our LSTM language modeling experiments on PTB, and on the role of each hyperparameter we tune.

We trained a 2-layer LSTM with 650 hidden units per layer and 650-dimensional word embeddings (similar to (Zaremba et al., 2014; Gal & Ghahramani, 2016)) on sequences of length 70 in mini-batches of size 40. To optimize the baseline LSTM, we used SGD with initial learning rate 30, which was decayed by a factor of 4 based on the non-monotonic criterion introduced by Merity et al. (2018) (i.e., whenever the validation perplexity fails to improve for 5 epochs). Following Merity et al. (2018), we used gradient clipping 0.25.

To optimize the ST-LSTM, we used the same optimization setup as for the baseline LSTM. For the hyperparameters, we used Adam with learning rate 0.01. We used an alternating training schedule in which we updated the model parameters for 2 steps on the training set and then updated the hyperparameters for 1 step on the validation set. We used one epoch of warm-up, in which we

updated the model parameters, but did not update hyperparameters. We terminated training when the learning rate dropped below 0.0003.

We tuned variational dropout (re-using the same dropout mask for each step in a sequence) on the input to the LSTM, the hidden state between the LSTM layers, and the output of the LSTM. We also tuned embedding dropout, which sets entire rows of the word embedding matrix to 0, effectively removing certain words from all sequences. We regularized the hidden-to-hidden weight matrix using DropConnect (zeroing out weights rather than activations) (Wan et al., 2013). Because DropConnect operates directly on the weights and not individually on the mini-batch elements, we cannot use independent perturbations per example; instead, we sample a single DropConnect rate per mini-batch. Finally, we used activation regularization (AR) and temporal activation regularization (TAR). AR penalizes large activations, and is defined as:

$$\alpha ||m \odot h_t||_2 \tag{63}$$

where $m$ is a dropout mask and $h_t$ is the output of the LSTM at time $t$. TAR is a slowness regularizer, defined as:

$$\beta ||h_t - h_{t+1}||_2 \tag{64}$$

For AR and TAR, we tuned the scaling coefficients $\alpha$ and $\beta$. For the baselines, the hyperparameter ranges were: $[0, 0.95]$ for the dropout rates, and $[0, 4]$ for $\alpha$ and $\beta$. For the ST-LSTM, all the dropout rates and the coefficients $\alpha$ and $\beta$ were initialized to 0.05 (except in Figure 3, where we varied the output dropout rate).

## E    IMAGE CLASSIFICATION EXPERIMENT DETAILS

Here, we present additional details on the CNN experiments. For all results, we held out 20% of the training data for validation.

We trained the baseline CNN using SGD with initial learning rate 0.01 and momentum 0.9, on mini-batches of size 128. We decay the learning rate by 10 each time the validation loss fails to decrease for 60 epochs, and end training if the learning rate falls below $10^{-5}$ or validation loss has not decreased for 75 epochs. For the baselines—grid search, random search, and Bayesian optimization—the search spaces for the hyperparameters were as follows: dropout rates were in the range $[0, 0.75]$; contrast, saturation, and brightness each had range $[0, 1]$; hue had range $[0, 0.5]$; the number of cutout holes had range $[0, 4]$, and the length of each cutout hole had range $[0, 24]$.

We trained the ST-CNN's elementary parameters using SGD with initial learning rate 0.01 and momentum of 0.9, on mini-batches of size 128 (identical to the baselines). We use the same decay schedule as the baseline model. The hyperparameters are optimized using Adam with learning rate 0.003. We alternate between training the best-response approximation and hyperparameters with the same schedule as the ST-LSTM, i.e. $T_{train} = 2$ steps on the training step and $T_{valid} = 1$ steps on the validation set. Similarly to the LSTM experiments, we used five epochs of warm-up for the model parameters, during which the hyperparameters are fixed. We used an entropy weight of $\tau = 0.001$ in the entropy regularized objective (Eq. 15). The cutout length was restricted to lie in $\{0, \ldots, 24\}$ while the number of cutout holes was restricted to lie in $\{0, \ldots, 4\}$. All dropout rates, as well as the continuous data augmentation noise parameters, are initialized to 0.05. The cutout length is initialized to 4, and the number of cutout holes is initialized to 1. Overall, we found the ST-CNN to be relatively robust to the initialization of hyperparameters, but starting with low regularization aided optimization in the first few epochs.

## F    ADDITIONAL DETAILS ON HYPERPARAMETER SCHEDULES

Here, we draw connections between hyperparameter schedules and curriculum learning. Curriculum learning (Bengio et al., 2009) is an instance of a family of *continuation methods* (Allgower & Georg, 2012), which optimize non-convex functions by solving a sequence of functions that are ordered by increasing difficulty. In a continuation method, one considers a family of training criteria $C_\lambda(\mathbf{w})$ with a parameter $\lambda$, where $C_1(\mathbf{w})$ is the final objective we wish to minimize, and $C_0(\mathbf{w})$ represents the training criterion for a simpler version of the problem. One starts by optimizing $C_0(\mathbf{w})$ and then gradually increases $\lambda$ from 0 to 1, while keeping $\mathbf{w}$ at a local minimum of $C_\lambda(\mathbf{w})$ (Bengio et al., 2009). This has been hypothesized to both aid optimization and improve generalization. In this

section, we explore how hyperparameter schedules implement a form of curriculum learning; for example, a schedule that increases dropout over time increases stochasticity, making the learning problem more difficult. We use the results of grid searches to understand the effects of different hyperparameter settings throughout training, and show that greedy hyperparameter schedules can outperform fixed hyperparameter values.

First, we performed a grid search over 20 values each of input and output dropout, and measured the validation perplexity in each epoch. Figure 7 shows the validation perplexity achieved by different combinations of input and output dropout, at various epochs during training. We see that at the start of training, the best validation loss is achieved with small values of both input and output dropout. As we train for more epochs, the best validation performance is achieved with larger dropout rates.

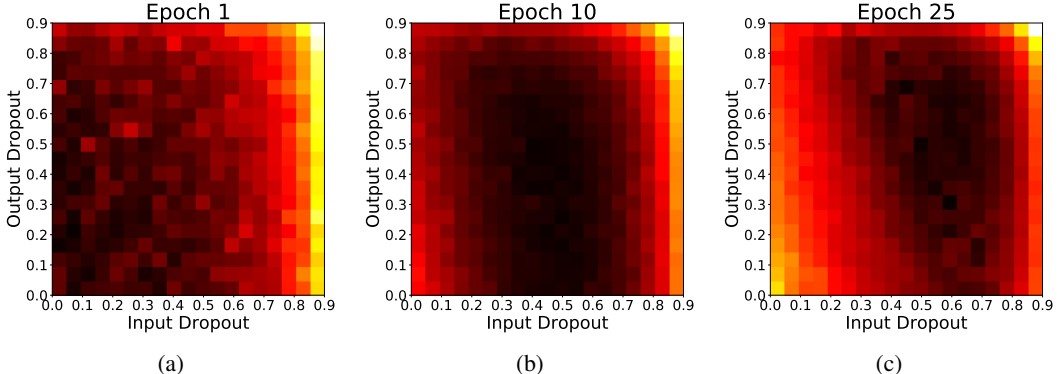

Figure 7: **Validation performance of a baseline LSTM given different settings of input and output dropout, at various epochs during training. (a)**, **(b)**, and **(c)** show the validation performance on PTB given different hyperparameter settings, at epochs 1, 10, and 25, respectively. Darker colors represent lower (better) validation perplexity.

Next, we present a simple example to show the potential benefits of greedy hyperparameter schedules. For a single hyperparameter—output dropout—we performed a fine-grained grid search and constructed a dropout schedule by using the hyperparameter values that achieve the best validation perplexity at each epoch in training. As shown in Figure 8, the schedule formed by taking the best output dropout value in each epoch yields better generalization than any of the fixed hyperparameter values from the initial grid search. In particular, by using small dropout values at the start of training, the schedule achieves a fast decrease in validation perplexity, and by using larger dropout later in training, it achieves better overall validation perplexity.

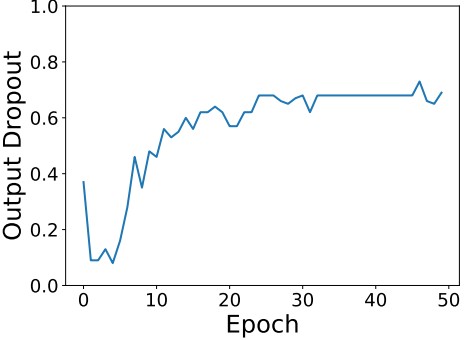

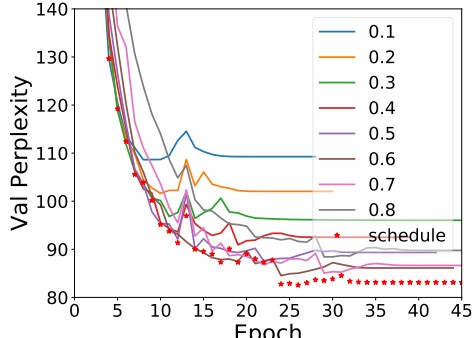

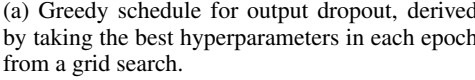

(a) Greedy schedule for output dropout, derived by taking the best hyperparameters in each epoch from a grid search.

(b) Comparison of fixed output dropout values and the dropout schedule derived from grid searches

Figure 8: **Grid search-derived schedule for output dropout.**

Figure 9 shows the perturbed values for output dropout we used to investigate whether the improved performance yielded by STNs is due to the regularization effect, and not the schedule, in Section 4.1.

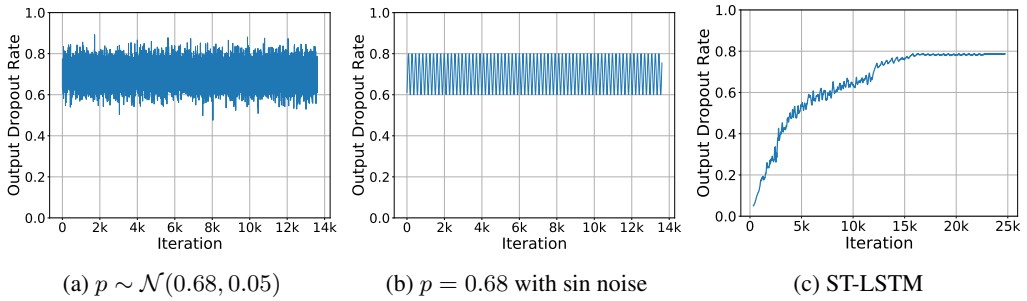

(a) $p \sim \mathcal{N}(0.68, 0.05)$      (b) $p = 0.68$ with sin noise      (c) ST-LSTM

Figure 9: **Comparison of output dropout schedules.** **(a)** Gaussian-perturbed output dropout rates around the best value found by grid search, 0.68; **(b)** sinusoid-perturbed output dropout rates with amplitude 0.1 and a period of 1200 mini-batches; **(c)** the output dropout schedule found by the ST-LSTM.

## G  CODE LISTINGS

In this section, we provide PyTorch code listings for the approximate best-response layers used to construct ST-LSTMs and ST-CNNs: the `HyperLinear` and `HyperConv2D` classes. We also provide a simplified version of the optimization steps used on the training set and validation set.

Listing 1: HyperLinear, used as a drop-in replacement for Linear modules

```python
class HyperLinear(nn.Module):

    def __init__(self, input_dim, output_dim, n_hparams):
        super(HyperLinear, self).__init__()

        self.input_dim = input_dim
        self.output_dim = output_dim
        self.n_hparams = n_hparams
        self.n_scalars = output_dim

        self.elem_w = nn.Parameter(torch.Tensor(output_dim, input_dim))
        self.elem_b = nn.Parameter(torch.Tensor(output_dim))

        self.hnet_w = nn.Parameter(torch.Tensor(output_dim, input_dim))
        self.hnet_b = nn.Parameter(torch.Tensor(output_dim))

        self.htensor_to_scalars = nn.Linear(self.n_hparams,
            self.n_scalars*2, bias=False)

        self.init_params()

    def forward(self, input, hnet_tensor):
        output = F.linear(input, self.elem_w, self.elem_b)

        if hnet_tensor is not None:
            hnet_scalars = self.htensor_to_scalars(hnet_tensor)
            hnet_wscalars = hnet_scalars[:, :self.n_scalars]
            hnet_bscalars = hnet_scalars[:, self.n_scalars:]

            hnet_out = hnet_wscalars * F.linear(input, self.hnet_w)
            hnet_out += hnet_bscalars * self.hnet_b

            output += hnet_out

        return output
```

Listing 2: HyperConv2d, used as a drop-in replacement for Conv2d modules

```python
class HyperConv2d(nn.Module):
```

```python
def __init__(self, in_channels, out_channels, kernel_size, padding,
    num_hparams,
    stride=1, bias=True):
    super(HyperConv2d, self).__init__()

    self.in_channels = in_channels
    self.out_channels = out_channels
    self.kernel_size = kernel_size
    self.padding = padding
    self.num_hparams = num_hparams
    self.stride = stride

    self.elem_weight = nn.Parameter(torch.Tensor(
        out_channels, in_channels, kernel_size, kernel_size))
    self.hnet_weight = nn.Parameter(torch.Tensor(
        out_channels, in_channels, kernel_size, kernel_size))
    if bias:
        self.elem_bias = nn.Parameter(torch.Tensor(out_channels))
        self.hnet_bias = nn.Parameter(torch.Tensor(out_channels))
    else:
        self.register_parameter('elem_bias', None)
        self.register_parameter('hnet_bias', None)

    self.htensor_to_scalars = nn.Linear(
        self.num_hparams, self.out_channels*2, bias=False)
    self.elem_scalar = nn.Parameter(torch.ones(1))

    self.init_params()

def forward(self, input, htensor):
    """
    Arguments:
        input (tensor): size should be (B, C, H, W)
        htensor (tensor): size should be (B, D)
    """
    output = F.conv2d(input, self.elem_weight, self.elem_bias,
        padding=self.padding,
        stride=self.stride)
    output *= self.elem_scalar
    if htensor is not None:
        hnet_scalars = self.htensor_to_scalars(htensor)
        hnet_wscalars = hnet_scalars[:,
            :self.out_channels].unsqueeze(2).unsqueeze(2)
        hnet_bscalars = hnet_scalars[:, self.out_channels:]

        hnet_out = F.conv2d(input, self.hnet_weight,
            padding=self.padding,
            stride=self.stride)
        hnet_out *= hnet_wscalars
        if self.hnet_bias is not None:
            hnet_out += (hnet_bscalars *
                self.hnet_bias).unsqueeze(2).unsqueeze(2)
        output += hnet_out
    return output

def init_params(self):
    n = self.in_channels * self.kernel_size * self.kernel_size
    stdv = 1. / math.sqrt(n)
    self.elem_weight.data.uniform_(-stdv, stdv)
    self.hnet_weight.data.uniform_(-stdv, stdv)
    if self.elem_bias is not None:
        self.elem_bias.data.uniform_(-stdv, stdv)
        self.hnet_bias.data.uniform_(-stdv, stdv)
```

```
      self.htensor_to_scalars.weight.data.normal_(std=0.01)
```

Listing 3: Stylized optimization step on the training set for updating elementary parameters

```
# Perturb hyperparameters around current value in unconstrained
# parametrization.
batch_htensor = perturb(htensor, hscale)

# Apply necessary reparametrization of hyperparameters.
hparam_tensor = hparam_transform(batch_htensor)

# Sets data augmentation hyperparameters in the data loader.
dataset.set_hparams(hparam_tensor)

# Get next batch of examples and apply any input transformation
# (e.g. input dropout) as dictated by the hyperparameters.
images, labels = next_batch(dataset)
images = apply_input_transform(images, hparam_tensor)

# Run everything through the model and do gradient descent.
pred = hyper_cnn(images, batch_htensor, hparam_tensor)
xentropy_loss = F.cross_entropy(pred, labels)
xentropy_loss.backward()
cnn_optimizer.step()
```

Listing 4: Stylized optimization step on the validation set for updating hyperparameters/noise scale

```
# Perturb hyperparameters around current value in unconstrained
# parametrization, so we can assess sensitivity of validation
# loss to the scale of the noise.
batch_htensor = perturb(htensor, hscale)

# Apply necessary reparametrization of hyperparameters.
hparam_tensor = hparam_transform(batch_htensor)

# Get next batch of examples and run through the model.
images, labels = next_batch(valid_dataset)
pred = hyper_cnn(images, batch_htensor, hparam_tensor)
xentropy_loss = F.cross_entropy(pred, labels)

# Add extra entropy weight term to loss.
entropy = compute_entropy(hscale)
loss = xentropy_loss - args.entropy_weight * entropy
loss.backward()

# Tune the hyperparameters.
hyper_optimizer.step()

# Tune the scale of the noise applied to hyperparameters.
scale_optimizer.step()
```

## H   SENSITIVITY STUDIES

In this section, we present experiments that show how sensitive our STN models are to different meta-parameters.

In particular, we investigate the effect of using alternative schedules (Figure 10) for the number of optimization steps performed on the training and validation sets.

Additionally, we investigate the effect of using different initial perturbation scales for the hyperparameters, which are either fixed or tuned (Figure 11).

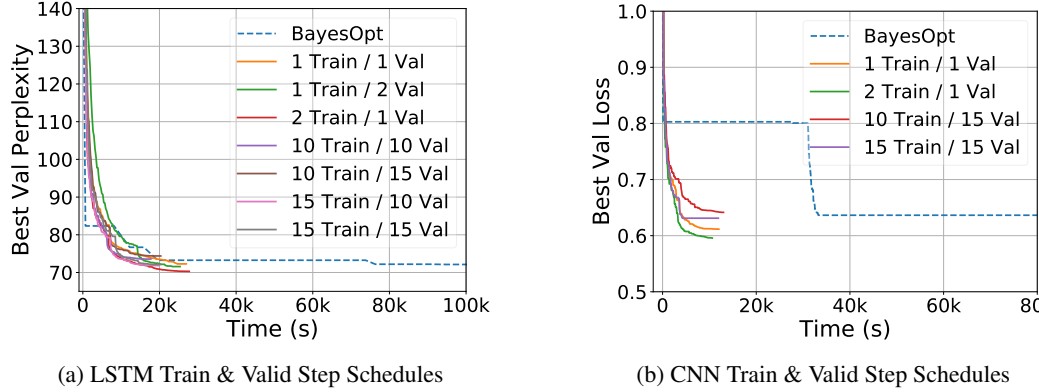

(a) LSTM Train & Valid Step Schedules     (b) CNN Train & Valid Step Schedules

Figure 10: **The effect of using a different number of train/val steps.** For the CNN, we include Bayesian Optimization and the reported STN parameters for comparison. During these experiments we found schedules which achieve better final loss with CNNs.

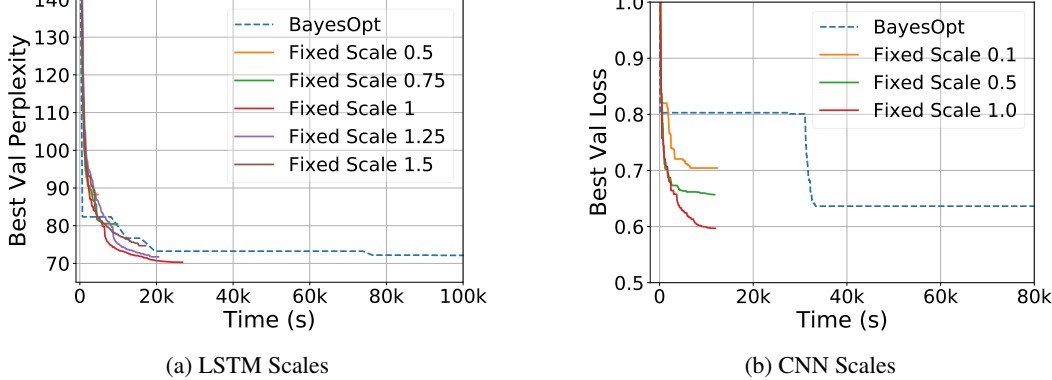

(a) LSTM Scales     (b) CNN Scales

Figure 11: **The effect of using different perturbation scales.** For the CNN, we include Bayesian Optimization and the reported STN parameters for comparison. For (a), the perturbation scales are fixed.

