# OpenReview forum: "Self-Tuning Networks: Bilevel Optimization of Hyperparameters using Structured Best-Response Functions"
_ICLR.cc/2019/Conference_

### Official Review · AnonReviewer3 · 2018-11-01
**The idea is interesing, but the explaination and experiment can be better**

**Rating:** 6
**Confidence:** 3

**Review:**

First, the writing can be better. I had a hard time to understand the paper. It has many symbols, but some of them are not explained. For instance, in  formula (9), what are Q or s? Also, formula (14). I probably can guess them. Is it possible to simplify the notations or use a table to list the symbols?

Finding good models is a bi-level or tri-level optimization problem. The paper describes a gradient-based hyperparameter optimization method, which finds model parameters, hyperparameter schedules, and network structure (limited) the same time. It is a interesting idea. Comparing random search, grid search and Spearmint, it seems to be better them. The paper rules out the performance gain is from the randomness of the hyperparameters, which is a good thought.

More evidences are needed to show this method is superior. The paper doesn't explain well why it works, and the experimental results are just ok. The network architecture search part is limited to number of filters in the experiments. Certainly, the results is not as good as  PNASNet or NASNet.

Evolution algorithm or GA shows good performance in hyperparameter optimization or neural architecture search. Why not compare with them? Random and grid search are not good generally, and Bayesian optimization is expensive and its performance depends on implementation.

In Table 2 and figure 4, should "Loss" be "Error"?

---

> ### Author Response · Authors · 2018-11-26
> **Response to Reviewer 3**
>
> Thank you for your feedback.
>
> Q: Clarity of writing
> First, the writing can be better. I had a hard time to understand the paper.
> It has many symbols, but some of them are not explained. For instance, in formula (9), what are Q or s? Also, formula (14). I probably can guess them. Is it possible to simplify the notations or use a table to list the symbols?
>
> A: We agree with the reviewer and appreciate the suggestions.  As such, we have added a table of notation to the appendix, along with simplifying our notation.
>
>
> Q: More evidences are needed to show this method is superior. The paper doesn't explain well why it works, and the experimental results are just ok. The network architecture search part is limited to number of filters in the experiments. Certainly, the results is not as good as  PNASNet or NASNet.
>
> A: STN techniques can be applied whenever there is a bilevel problem in which the lower-level variables are a neural network's parameters.  Therefore, STNs could be used for neural architecture search. However, they are most valuable when the upper-level variable does not affect the upper-level objective directly.  This is usually the case with regularization hyperparameters since one can not use naive simultaneous gradient descent (as in ENAS[3]). Thus, the focus in this paper is on regularization hyperparameters, separate from the network topology.
>
> We are interested in using STNs for jointly tuning regularization hyperparameters and network topology in future work. There may be benefits over current approaches since they either do not attempt to approximate the response gradient (see Eq. 6) like ENAS[3] or approximate it using finite differences like DARTS[4]. Directly approximating the best-response and including its gradient may yield a more accurate gradient of the validation loss.
>
> [1] Zoph, Barret, and Quoc V. Le. "Neural architecture search with reinforcement learning." arXiv preprint arXiv:1611.01578(2016).
> [2] Liu, Chenxi, et al. "Progressive neural architecture search." arXiv preprint arXiv:1712.00559 (2017).
> [3] Pham, Hieu, et al. "Efficient Neural Architecture Search via Parameter Sharing." arXiv preprint arXiv:1802.03268 (2018).
> [4] Liu, Hanxiao, Karen Simonyan, and Yiming Yang. "Darts: Differentiable architecture search." arXiv preprint arXiv:1806.09055 (2018).
>
>
> Q: Evolution algorithm or GA shows good performance in hyperparameter optimization or neural architecture search. Why not compare with them? Random and grid search are not good generally, and Bayesian optimization is expensive and its performance depends on implementation.
>
> A: We include a comparison to Hyperband for LSTMs in the revised version of the paper.
>
>
> Q: In Table 2 and figure 4, should "Loss" be "Error"?
>
> A: Figure 4 (now 5) is the loss because that is the objective being minimized via gradient descent.

---

> > ### Comment · AnonReviewer3 · 2018-11-27
> > **Response to the comments of authors**
> >
> > The notation table definitely helps. Ideally, I'd like to see that the Hyperband method is used in all experiments.

---

### Official Review · AnonReviewer1 · 2018-11-02
**Good idea, not clear if it is easy to apply.**

**Rating:** 6
**Confidence:** 3

**Review:**


========\\
Summary\\
========\\

The paper deals with hyper-parameter optimization of neural networks. The authors formulate the problem as a bilevel optimization problem: minimizing the validation loss over the hyperparameters, subject to the parameters being at the minimum of the training loss. The authors propose an approximation of the so-called best-response function, that maps the hyperparameters to the corresponding optimal parameters (w.r.t the minimization of the training loss), allowing a formulate as a single-level optimization problem and the use gradient descent algorithm. The proposed
approximation is based on shifting and scaling the weights and biases of the network. There are no guarantee on its quality except in some very simple cases. The approach assumes a distribution on the hyperparameters, governed by a parameter, which is adapted during the course of the training to achieve a compromise between the flexibility of the best-response function and the quality of its local approximation around the current hyperparameters. The authors show
that their approach beats grid-search, random search and Bayesian optimization on the CIFAR-10 and PTB datasets. They point out that the dynamic update of the hyperparameters during the training allows to reach a better performance than any fixed hyperparameter. \\


======================\\
Comments and questions\\
======================\\

Can cross-validation be adapted to this approach? \\

Can this be used to optimize the learning rate? Which is of course a crucial hyperparameter and that needs an update schedule during the training. \\

Section 3.2:\\

"If the entries are too large, then θ̂ φ will not be flexible enough to capture the best- response over the sampled neighborhood. However, its entries must remain sufficiently large so that θ̂ φ captures the local shape around the current hyperparameter values." Not clear why -- more explanations would be helpful. \\

"minimizing the first term eventually moves all probability mass towards an optimum λ∗ ,resulting in σ = 0". I can't see how minimizing the first term w.r.t \phi (as in section "2.2.Local approximation") would alter \sigma. \\

"τ must be set carefully to ensure...". The authors still do not explain how to set \tau. \\

Section 3.3: \\

If the hyperparameter is discrete and falls in Case 2, then REINFORCE gradient estimator is used. What about the quality of this gradient? \\

Section 5, paragraph Gradient-Based HO: "differentiating gradient descent" needs reformulation -- an algorithm cannot be differentiated. \\

Pros \\
- The paper is pretty clear \\
- Generalizes a previous idea and makes it handle discrete hyperparameters and scale better. \\
- I like the idea of hyperparameters changing dynamically during the training which allows to explore a much larger space than one value \\
- Although limited, the experimental results are convincing \\

Cons \\
- The method itself depends on some parameters and it is not clear how to choose them. Therefore it might be tricky to make it work in practice. I feel like there is a lot of literature around HO but very often people still use the very simple grid/random search, because the alternative methods are often quite complex to implement and make really work. So the fact that the method depends on "crucial" parameters but that are not transparently managed may be a big drawback to its applicability. \\
- No theoretical guarantee on the quality of the used approximation for neural networks \\
- Does not handle the learning rate which is a crucial hyperparameter (but maybe it could) \\

---

> ### Author Response · Authors · 2018-11-26
> **Response to Reviewer 2**
>
> Thank you for your feedback.
>
> Q: Can cross-validation be adapted to this approach?
>
> A: Yes, this approach can be adapted to k-fold cross-validation. The outer objective would be the sum of the validation losses across all the folds.  K different best response approximations would be trained on the k training sets, using a shared distribution over the hyperparameters.
>
>
> Q: Can this be used to optimize the learning rate? Which is of course a crucial hyperparameter and that needs an update schedule during the training.
>
> A: No, the learning rate cannot be optimized using this approach. Learning rates are hyperparameters of our optimization algorithm for solving the bilevel program, but the algorithm for solving the program is separate from the program itself. Hence, learning rates are not part of the program’s variables. However, it is straightforward to combine STNs with existing methods for optimizing learning rates in the literature [1,2,3].
>
> [1] Nicol N Schraudolph. Local gain adaptation in stochastic gradient descent. International Conference on Artificial Neural Networks (ICANN), 1999.
> [2] Tom Schaul, Sixin Zhang, and Yann LeCun. No more pesky learning rates. International Conference on Machine Learning (ICML), 2013.
> [3] Atilim G Baydin, Robert Cornish, David Martinez Rubio, Mark Schmidt, and Frank Wood. Online learning rate adaptation with hypergradient descent. International Conference on Learning Representations (ICLR), 2018.
>
>
> Q: Section 3.2: "If the entries are too large, then θ̂ φ will not be flexible enough to capture the best- response over the sampled neighborhood. However, its entries must remain sufficiently large so that θ̂ φ captures the local shape around the current hyperparameter values." Not clear why -- more explanations would be helpful.
>
> A: If we sample hyperparameters over a large range where the best-response is highly non-linear, we will never learn a good linear approximation. If we sample hyperparameters from too small a range - say a point mass - then the approximation learned will only be valid at that single point, and will not give a correct gradient.  We must sample over a range where the best-response is approximately linear. In other words, the range of the region sampled should match the flexibility of the best-response approximation. We have added Figure 2 in the paper to help clarify this.
>
>
> Q: "minimizing the first term eventually moves all probability mass towards an optimum λ∗ ,resulting in σ = 0". I can't see how minimizing the first term w.r.t \phi (as in section "2.2.Local approximation") would alter \sigma.
>
> A: The objective \hat{F}_V is optimized w.r.t. \lambda and \sigma, not \phi. In general, for any function g with minimum \lambda*, minimizing E_{p(\eps|\sigma)}[g(\lambda + \eps)] w.r.t. \lambda and \sigma will achieve an optimum at (\lambda*, 0), since sampling a nonzero \eps when at \lambda=\lambda* will cause the expectation to increase in value.
>
>
> Q: "τ must be set carefully to ensure...". The authors still do not explain how to set \tau.
>
> A: We set \tau by doing a grid-search. However, we found a default value of tau=0 to work well across our experiments.
>
>
> Q: Section 3.3: If the hyperparameter is discrete and falls in Case 2, then REINFORCE gradient estimator is used. What about the quality of this gradient?
>
> A: We found it to work well empirically for tuning the number of hidden units. If variance grew too high, it would be possible to use various variance reduction techniques such as RELAX [1].
>
> [1] Grathwohl, Will, Choi, Dami, Wu, Yuhuai, Roeder, Geoff, Duvenaud, David. “Backpropagation through the Void: Optimizing control variates for black-box gradient estimation”. ICLR 2018

---

> > ### Author Response · Authors · 2018-11-26
> > **Response to Review 2 Continued**
> >
> > Q: Section 5, paragraph Gradient-Based HO: "differentiating gradient descent" needs reformulation -- an algorithm cannot be differentiated.
> >
> > A: We have removed this terminology from the paper. To clarify, we view gradient descent as a function grad_descent(initial_weight, optimizer_parameters, hyperparameters) which returns final_weight as in [1]. This descent function is differentiable w.r.t. the hyperparameters as long as the hyperparameters are not discrete and the training loss is differentiable w.r.t. the hyperparameters.
> >
> > [1] Maclaurin, Dougal, David Duvenaud, and Ryan Adams. "Gradient-based hyperparameter optimization through reversible learning." International Conference on Machine Learning. 2015.
> >
> >
> > Q: Cons - The method itself depends on some parameters and it is not clear how to choose them. Therefore it might be tricky to make it work in practice. I feel like there is a lot of literature around HO but very often people still use the very simple grid/random search, because the alternative methods are often quite complex to implement and make really work. So the fact that the method depends on "crucial" parameters but that are not transparently managed may be a big drawback to its applicability.
> >
> > A: For implementation, it is easy to apply STN techniques to existing deep learning models by replacing existing modules with “hyper” versions which take an additional vector of hyperparameters in addition to the usual input. These hyper-modules are precisely the approximate best response functions in Equation 13. We include code listings in the appendix showing our HyperLinear and HyperConv2d modules.
> >
> > The STN training algorithm has a few metaparameters, including the schedule for the number of gradient steps performed on the training and validation sets, the initializations, and the learning rates for the base parameter optimizer and the hyperparameter optimizer. We found that default values for each of these hyperparameters work well across the tasks we investigated - these are included in Table 4.
> >
> > Additionally, we have included ablation studies of the metaparameters in section H of the appendix.  These show how robust the optimization procedure is to the meta-parameters, and the importance of the response gradient.
> >
> > Thus, STNs are easy to apply with minimal manual tuning by using the default configuration.
> >
> >
> > Q: - No theoretical guarantee on the quality of the used approximation for neural networks
> >
> > A: While it is true there are no theoretical guarantees on the quality of the approximation, it is common to lack theoretical guarantees when developing new algorithms for neural networks. Indeed, it is an active area of research to prove the convergence of gradient descent even in shallow, nonlinear networks [1][2][3]. Incorporating the bilevel structure of the problem will likely introduce additional complications, although we hope to investigate this further in future work.
> >
> > [1] Difan Zou, Yuan Cao, Dongruo Zhou, and Quanquan Gu. “Stochastic Gradient Descent Optimizes Over-parameterized Deep ReLU Networks”. Preprint, 2018.
> > [2] Simon S. Du, Jason D. Lee, Haochuan Li, Liwei Wang, and Xiyu Zhai. “Gradient Descent Finds Global Minima of Deep Neural Networks”. Preprint, 2018.
> > [3] Zeyuan Allen-Zhu, Yuanzhi Li, and Zhao Song. “A Convergence Theory for Deep Learning via Over-Parametrization”. Preprint, 2018.

---

### Official Review · AnonReviewer2 · 2018-11-03
**Principled approach to hyperparameter tuning but only evaluated on small scale problems to-date.**

**Rating:** 7
**Confidence:** 4

**Review:**

The paper proposes a bilevel optimization approach for hyperparameter tuning. This idea is not new having been proposed in works prior to the current resurgence of deep learning (e.g., Do et al., 2007, Domke 2012, and Kunisch & Pock, 2013). However, the combination of bilevel optimization for hyperparameter tuning with approximation is interesting. Moreover, the proposed approach readily handles discrete parameters.

Experiments are run on small scale problems, namely, CIFAR-10 and PTB. Results are encouraging but not stellar. More work would need to be done to validate the utility of the proposed approach on larger scale problems.

---

> ### Author Response · Authors · 2018-11-26
> **Response to Reviewer 1**
>
> Thank you for your feedback.
>
> Q: Experiments are run on small scale problems, namely, CIFAR-10 and PTB. Results are encouraging but not stellar. More work would need to be done to validate the utility of the proposed approach on larger scale problems.
>
> A: Smaller datasets such as CIFAR-10 and PTB provide an ideal testbed for hyperparameter optimization algorithms since performance depends heavily on regularization. The architectures used for RNNs are comparable in size to top-performing architectures on PTB [1,2]. In addition, we believe we are the first to tune RNN hyperparameters using gradient-based methods since these hyperparameters are often dropout probabilities that other gradient-based methods can’t handle.
>
> AlexNet is a standard architecture used when ResNets are too powerful and can overfit. This convolutional architecture is comparable to the largest tuned via gradient-based hyperparameter optimization methods in the literature. Papers such as [3,4] evaluate their algorithms on MNIST-size datasets using logistic regression or small feed-forward networks. In [5] a similar size convolutional network to AlexNet is tuned, but they weren’t able to tune data augmentation hyperparameters and had to use continuous dropout noise to obtain a gradient, unlike our method.
>
>
> [1] Merity, Stephen, Keskar, Nitish S., and Socher, Richard. "Regularizing and optimizing LSTM language models." ICLR 2018.
> [2] Melis, Gabor, Dyer, Chris, and Blunsom, Phil. “On the State of the Art of Evaluation in Neural Language Models” ICLR 2018
> [3] Pedregosa, Fabian. “Hyperparameter optimization with approximate gradient” ICML 2016
> [4] Maclaurin, Dougal, Duvenaud, David, and Adams, Ryan. “Gradient-based Hyperparameter Optimization through Reversible Learning” ICML 2015
> [5] Luketina, Jelena, Berglund, Mathias, Greff, Klaus, and Raiko, Tapani. “Scalable Gradient-Based Tuning of Continuous Regularization Hyperparameters” ICML 2016

---

### Author Response · Authors · 2018-11-27
**Summary of changes**

We thank all the reviewers for their helpful comments.

We have made the following changes to the paper to address reviewer concerns:

--- Improved clarity: We simplified our notation and included a table of notation in Appendix A. We added an additional figure which clarifies why hyperparameters must be sampled carefully. We have also included a discussion of the direct/response gradient which clarifies our approach.

--- Sensitivity to metaparameters: In response to concerns about the sensitivity of our algorithm to its “metaparameters”, we have included sensitivity studies in Appendix H to examine how our method performs under various metaparameter settings.

--- Ease of implementation: We emphasize that STNs are easy to implement and use in code simply by replacing existing deep learning modules with “hyper” counterparts. To illustrate this, we added code listings used for our experiments in Appendix G.

--- Comparison to additional hyperparameter optimization methods: We have included a comparison to Hyperband for our LSTM experiments.

---

### Meta-Review · Area_Chair1 · 2018-12-19
**A useful approach to hyperparameter tuning, promising results**

**Confidence:** 4
**Recommendation:** Accept (Poster)

**Metareview:**

The paper proposes an approach to hyperparameter tuning based on bilevel optimization, and demonstrates promising empirical results. Reviewer's concerns seem to be addressed well in rebuttals and extended version of the paper.